# Mps2 links Csm4 and Mps3 to form a telomere-associated LINC complex in budding yeast

Jinbo Fan, Hui Jin, Bailey A Koch, Hong-Guo Yu ⓘ

The linker of the nucleoskeleton and cytoskeleton (LINC) complex is composed of two transmembrane proteins: the KASH domain protein localized to the outer nuclear membrane and the SUN domain protein to the inner nuclear membrane. In budding yeast, the sole SUN domain protein, Mps3, is thought to pair with either Csm4 or Mps2, two KASH-like proteins, to form two separate LINC complexes. Here, we show that Mps2 mediates the interaction between Csm4 and Mps3 to form a heterotrimeric telomere-associated LINC (t-LINC) complex in budding yeast meiosis. Mps2 binds to Csm4 and Mps3, and all three are localized to the telomere. Telomeric localization of Csm4 depends on both Mps2 and Mps3; in contrast, Mps2's localization depends on Mps3 but not Csm4. Mps2-mediated t-LINC complex regulates telomere movement and meiotic recombination. By ectopically expressing *CSM4* in vegetative yeast cells, we reconstitute the heterotrimeric t-LINC complex and demonstrate its ability to tether telomeres. Our findings therefore reveal the heterotrimeric composition of the t-LINC complex in budding yeast and have implications for understanding variant LINC complex formation.

## Introduction

The linker of the nucleoskeleton and cytoskeleton (LINC) complex has emerged as a key regulator for a diverse range of nuclear activities that include chromosome movement, nuclear positioning, and gene expression (Tapley & Starr, 2013; Chang et al, 2015; Burke, 2018). The canonical LINC complex is composed of two transmembrane proteins, the SUN (Sad1 and UNC-84) protein localized to the inner nuclear membrane (INM) and the KASH (Klarsicht, ANC-1 and Syne/Nesprin homology) protein localized to the outer nuclear membrane (ONM) (Starr et al, 2001; Crisp et al, 2006). The canonical SUN-KASH interaction takes place primarily in the perinuclear space; therefore, the LINC complex not only bridges the INM and ONM but also connects the cytoskeleton to the nucleoskeleton and chromatin, allowing transduction of mechanical forces through the nuclear envelope (Starr & Fridolfsson, 2010). At least five SUN domain proteins and six KASH domain proteins have been found in mammals (Rajgor & Shanahan, 2013; Duong et al, 2014; Chang et al, 2015; Nishioka et al, 2016), potentially forming a diverse number of LINC complex variants. Similarly, SUN- and KASH-like proteins are prevalent in land plants (Zhou & Meier, 2013; Gumber et al, 2019). LINC proteins are believed to form hetero-dimeric hexamers and possibly higher ordered protein arrays for force transmission (Luxton et al, 2010; Sosa et al, 2012; Wang et al, 2012). How they are assembled in vivo to carry out diverse functions remains to be further determined.

In budding yeast, Mps3 is the sole SUN domain protein, which is concentrated at the yeast centrosome (Jaspersen et al, 2002), often called the spindle pole body (SPB). Mps3 also localizes to the INM (Jaspersen et al, 2002). Budding yeast lacks a canonical KASH domain protein, but possesses two ONM-localized KASH-like proteins: Mps2, present in both mitosis and meiosis (Winey et al, 1991), and Csm4, a meiosis-specific protein (Conrad et al, 2008; Kosaka et al, 2008; Koszul et al, 2008; Wanat et al, 2008). The genes encoding Mps2 and Csm4 are considered paralogs, but they differ drastically in protein size and show limited similarity at the amino acid level (Fig 1A). The current notion posits that in mitosis and presumably also in meiosis, Mps3 pairs with Mps2 at the centrosome to form a centrosome-associated LINC complex (Jaspersen et al, 2006; Chen et al, 2019), to which we refer as the c-LINC complex. In meiosis, Mps3 pairs with Csm4 to form a telomere-associated LINC (t-LINC) complex (Conrad et al, 2008; Kosaka et al, 2008; Koszul et al, 2008; Wanat et al, 2008).

Classified as single-pass type-II transmembrane proteins, Mps2 and Mps3 most likely interact in the perinuclear space through their corresponding C-terminal KASH-like and SUN domains, about 60 and 190 amino acids in size, respectively (Fig 1A), although an exception of their interaction at the centrosome has been reported (Chen et al, 2019). Like Mps3, Mps2 is also concentrated at the SPB, where it binds to additional SPB components to form a subcomplex that regulates SPB insertion into the nuclear envelope (Munoz-Centeno et al, 1999; Schramm et al, 2000). In vegetative yeast cells, Mps3 plays additional roles in DNA double-strand break repair by anchoring chromosomes to the nuclear periphery (Kalocsay et al,

Department of Biological Science, Florida State University, Tallahassee, FL, USA

Correspondence: hyu@fsu.edu
Jinbo Fan's present address is The Molecular Virology and Viral Immunology Laboratory, Xi'an Medical University, Xi'an, China

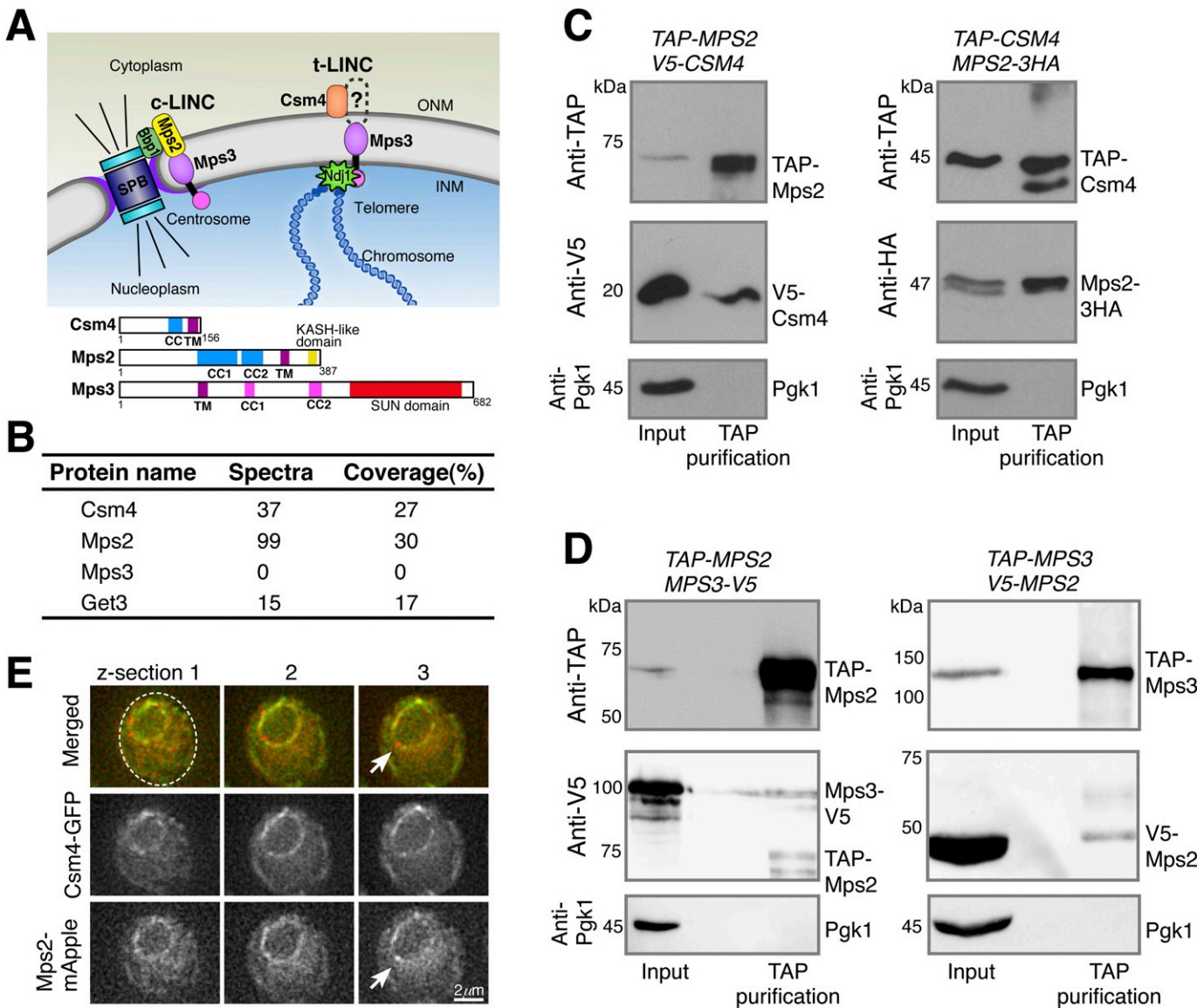

**Figure 1. Meiotic Mps2 binds to Csm4 and Mps3.**
**(A)** Schematic diagram showing the composition and location of c-LINC and t-LINC complexes in budding yeast. Domain organization of Csm4, Mps2, and Mps3 is shown at the bottom. **(B)** List of representative proteins copurified with TAP-Csm4. **(C)** Reciprocal immunoprecipitation showing Mps2-Csm4 interaction. The level of Pgk1 serves as a negative control for affinity purification. **(D)** Reciprocal immunoprecipitation showing Mps2-Mps3 interaction. Note that the anti-V5 antibody also recognizes TAP-Mps2. At least two biological replicates were performed. **(E)** Localization of Mps2 and Csm4 at prophase I. Three continuous optical sections are shown. Arrows point to the putative localization of Mps2 to the spindle pole body. Note that both Csm4 and Mps2 localize to the nuclear periphery. Dashed oval shows the overall cell shape. Red, Mps2-mApple; green, Csm4-GFP. CC, coiled coil; INM, inner nuclear membrane; ONM, outer nuclear membrane; TM, transmembrane domain.

2009; Oza et al, 2009). Whether Mps2 has a similar role in recombination and chromosome tethering remains unclear.

During meiosis, Mps3 is required for tethering telomeres to the nuclear envelope, in addition to its roles in SPB duplication and separation (Conrad et al, 2007, 2008; Li et al, 2015, 2017). At meiotic prophase I, the N terminus of Mps3, which is located in the nucleoplasm, binds to Ndj1, a telomere-associated protein (Chua & Roeder, 1997; Conrad et al, 1997), whereas its C-terminal SUN domain, located in the perinuclear space, has been proposed to bind to Csm4 (Fig 1A and Conrad et al, 2008). Therefore, the t-LINC complex, composed of Csm4 and Mps3 at a minimum, is capable of linking telomeres to the cytoplasmic actin filaments, which provide the mechanical forces necessary for meiotic telomere movement (Koszul et al, 2008). This t-LINC–dependent motility mediates the configuration of the telomere bouquet and can drastically deform the nucleus at prophase I (Conrad et al, 2008; Koszul et al, 2008). However, Csm4 is a tail-anchored membrane protein with its predicted transmembrane domain located at the very end of the C terminus, exposing merely a three-amino acid tail in the perinuclear space (Fig 1A). For canonical KASH proteins, their tails are usually ~30 amino-acid long (Sosa et al, 2012; Wang et al, 2012). In the absence of a KASH-like domain as found in Mps2,

how Csm4 interacts with Mps3 in the perinuclear space remains unclear.

We hypothesize that another unidentified factor is required to mediate Csm4 and Mps3 interaction to form the t-LINC complex. We report here that Mps2 mediates the interaction between Csm4 and Mps3 to form a heterotrimeric t-LINC complex that tethers telomeres and regulates nuclear dynamics in budding yeast meiosis. Using a combined cytological and genetic approach, we show that Mps2 is a major binding partner of Csm4 and a telomere-associated protein. Furthermore, by ectopically expressing *CSM4* in vegetative yeast cells, we have reconstituted the heterotrimeric t-LINC complex and demonstrated its ability to tether telomeres. Our findings, therefore, reveal the heterotrimeric composition of the yeast t-LINC complex.

# Results

### Meiotic Mps2 is a major binding partner of Csm4

To test our hypothesis that an unidentified factor mediates the interaction between Csm4 and Mps3, we performed TAP-Csm4 protein affinity purification, followed by mass spectrometry–based protein identification (Fig 1B). Purification of Csm4 was confirmed by 37 identified peptide spectra that belonged to Csm4 and covered 27% of its amino acid sequence (Fig 1B). As expected, Get3, which is required for insertion of tail-anchored proteins into the ER membrane (Schuldiner et al, 2008), was copurified with TAP-Csm4 (Fig 1B). However, Mps3 was not identified (Fig 1B). Unexpectedly, a major protein copurified with TAP-Csm4 was Mps2, which showed 99 peptide spectra that covered 30% of the Mps2 protein sequence (Fig 1B). This and additional findings described below prompted us to propose that Mps2 is a t-LINC component in budding yeast.

To further determine the interaction between t-LINC components, we performed reciprocal immunoprecipitation, followed by Western blotting (Fig 1C and D). Yeast cells were induced to undergo synchronous meiosis and arrested at prophase I by way of *ndt80Δ* (Xu et al, 1995). V5-Csm4 was copurified with TAP-Mps2; reciprocally, Mps2-3HA was purified by immunoprecipitation of TAP-Csm4 (Fig 1C). By fluorescence microscopy, we confirmed that Mps2 and Csm4 colocalized to the nuclear periphery during meiosis (Fig 1E). In addition, Mps3-V5 was copurified with TAP-Mps2; V5-Mps2 was copurified with TAP-Mps3 (Fig 1D). With this TAP method, however, we did not observe a physical interaction between Csm4 and Mps3 (Fig 1B and our unpublished data), indicating that either Csm4 interacts indirectly with Mps3, or their interaction is weak. In summary, we have revealed the interaction between Mps2 and Csm4 and confirmed the interaction between meiotic Mps2 and Mps3.

### Mps2 is required for meiotic cell progression

To better understand the role of Mps2 in meiosis, first, we determined Mps2 localization by time-lapse fluorescence microscopy (Fig 2A). As expected, meiotic Mps2 was found at the SPB, evidenced by its colocalization with the SPB marker Tub4, which first appeared

as a focus, then separated from one focus to two foci in meiosis I and from two to four in meiosis II (Figs 2A and S1A). In early meiosis II, Mps2 was preferentially associated with the newly duplicated SPB, which displayed a weaker Tub4-mApple signal (Fig 2A, t = 70) likely because of a slower fluorescence maturation time of mApple than that of GFP. Importantly, Mps2 localized around the nuclear periphery (Figs 1E, 2A, and S2A). Before SPB separation in meiosis I, distribution of Mps2 at the nuclear envelope appeared uneven, at times displaying high occupancy to only half or less than half of the nuclear periphery (Fig 2A, t = −30 min for an example and Fig S2A). This polarized localization of Mps2 to a certain region of the nuclear envelope is similar to that of Csm4 at prophase I (Kosaka et al, 2008; Wanat et al, 2008) and lends support to the idea that like Csm4, Mps2 is a component of the t-LINC complex.

Next, we generated a meiosis-specific Mps2-depletion allele, $P_{CLB2}$-*MPS2*, in which the endogenous *MPS2* promoter was replaced by that of *CLB2*, halting production of Mps2 at the onset of meiosis (Fig 2B and [Lee & Amon, 2003]). During meiosis, the level of Mps2 increased 4 h after the induction, which roughly corresponded to meiosis I. On the other hand, depletion of meiotic Mps2 appeared to be near completion in $P_{CLB2}$-*MPS2* cells (Fig 2B). In the absence of Mps2, meiosis I occurred in more than 60% of the cells, as determined by the separation of Tub4-mApple from one focus to two foci (Figs 2C and S1A–E). More than 80% of wild-type cells completed meiosis, forming four Tub4-mApple foci; in contrast, less than 5% of $P_{CLB2}$-*MPS2* cells displayed three or four Tub4-mApple foci (Figs 2C and S1B and D), indicating that Mps2 is required for meiotic cell progression. Abolishing meiotic recombination by way of *spo11-Y135F* (Keeney et al, 1997) did not rescue the SPB separation defect in $P_{CLB2}$-*MPS2* cells; however, the double-mutant *spo11-Y135F* $P_{CLB2}$-*MPS2* completed SPB separation in meiosis I earlier than both the wild-type and $P_{CLB2}$-*MPS2* cells (Fig S1C and E). In comparison, two rounds of SPB separation occurred in *csm4Δ* cells, but there was a delay in meiosis I as in $P_{CLB2}$-*MPS2* cells (Fig S1F). The phenotype of delayed SPB separation in *csm4Δ* was also suppressed by *spo11-Y135F* (Fig S1F and G). Together, these findings support the idea that Mps2 plays a role in meiotic recombination as Csm4 does (Kosaka et al, 2008; Koszul et al, 2008; Wanat et al, 2008). In addition, meiotic Mps2 has independent functions outside of the t-LINC complex, and is likely involved in c-LINC–mediated meiotic SPB duplication as it is in mitosis, a topic for future study.

### Mps2 is required for nuclear localization of Csm4 but not for Mps3

To further test our hypothesis that Mps2 is a component of the t-LINC complex, we determined the interdependency of t-LINC components for their roles in nuclear envelope localization. We focused on meiotic cells at prophase I when the t-LINC complex is active in regulating telomere bouquet formation and meiotic recombination (Conrad et al, 2008; Kosaka et al, 2008; Koszul et al, 2008; Wanat et al, 2008). By fluorescence microscopy, we observed that Mps2 and Mps3 colocalized to both the nuclear periphery and the SPB (Fig 2D). Similarly, their localization to the nuclear periphery appeared uneven at prophase I (Fig 2D). We note that some Mps3 foci at the nuclear periphery did not colocalize with Mps2, indicating that Mps3 forms protein aggregates outside of the context of the LINC complex (our unpublished data). Because the t-LINC

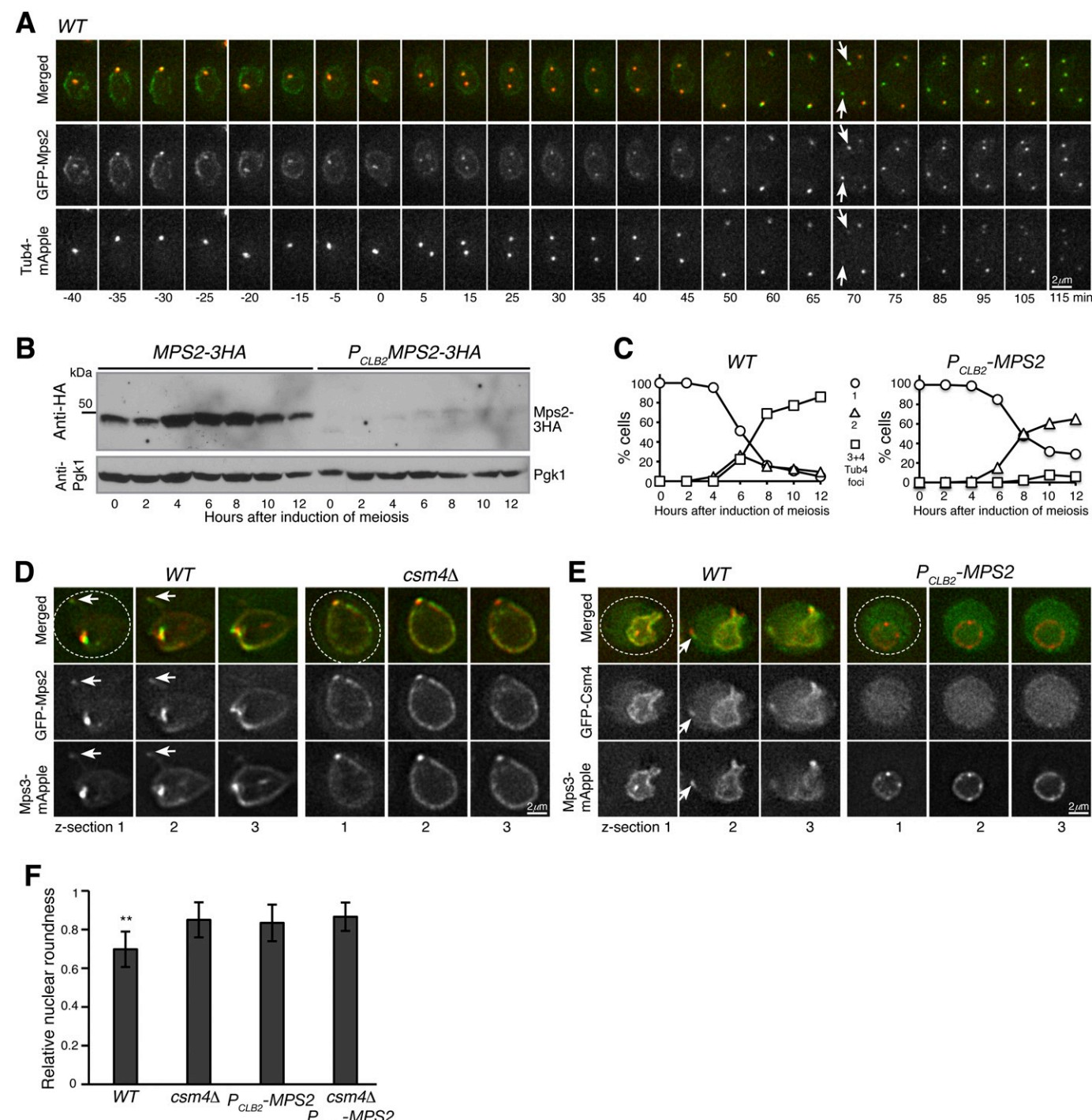

**Figure 2. Mps2 is required for meiotic cell progression and regulates Csm4 localization.**
**(A)** Time-lapse fluorescence microscopy showing GFP-Mps2 localization during meiosis. Tub4-mApple serves as a marker for the spindle pole body (SPB). Projected images of 12 z-sections are shown. Arrows point to the newly duplicated SPBs in meiosis II. Time in minutes is shown at the bottom. Time zero refers to the onset of SPB separation in meiosis I. Note the uneven localization of Mps2 to the nuclear periphery in meiosis I. Red, Tub4-mApple; green, GFP-Mps2. **(B)** Protein level of Mps2 in budding yeast meiosis. Yeast cells were induced to undergo synchronous meiosis; cell aliquots were withdrawn at indicated times. The level of Mps2-3HA was probed by an anti-HA antibody. The level of Pgk1 serves as a loading control. Note that Mps2 was largely depleted in $P_{CLB2}$-MPS2 cells. **(C)** SPB separation in wild-type (WT) and $P_{CLB2}$-MPS2 cells during meiosis. Tub4-mApple serves as a marker for the SPB. At least 100 cells were counted at each time point. Three biological replicates were performed, shown is a representative. Note that $P_{CLB2}$-MPS2 cells were stopped with only two SPBs. **(D)** Colocalization of Mps2 and Mps3. Note that GFP-Mps2 (green) and Mps3-mApple (red) remain bound to the nuclear periphery in the csm4Δ cell. **(E)** Mps2 is required for nuclear localization of Csm4. Note that the nucleus becomes a sphere in the $P_{CLB2}$-MPS2 cell. Three continuous optical sections are shown in (D, E). Arrows point to the nuclear protrusion. Dashed ovals show the overall cell shape. Red, Mps3-mApple; green, GFP-Csm4. **(F)** Quantification of nuclear shape at prophase I. Nuclear roundness was determined in WT, csm4Δ, $P_{CLB2}$-MPS2, and csm4Δ $P_{CLB2}$-MPS2 cells. A perfect sphere is defined as roundness factor 1. Single optical sections were used to trace the nuclear periphery. Cells were arrested at prophase I by way of ndt80Δ. Significant difference ($P < 0.01$) is indicated as **. At least 100 cells were counted from each strain.

complex mediates actin-based motility at the nuclear periphery (Koszul et al, 2008), the nuclear envelope formed membrane protrusions, and the nuclear shape became highly irregular at prophase I (Fig 2D and E and [Conrad et al, 2008; Koszul et al, 2008]). As expected, Mps2, Mps3, and Csm4 all were present at the leading edge of the nuclear protrusions (Fig 2D and E, arrows). In the absence of Csm4, the meiotic nucleus appeared as a sphere and lacked nuclear protrusions (Fig 2D and [Conrad et al, 2008; Koszul et al, 2008]). Mps2 and Mps3 remained localized to, but were distributed evenly around, the nuclear periphery in csm4Δ cells, demonstrating that association of both Mps2 and Mps3 with the nuclear envelope is independent of Csm4, but their polarized nuclear localization depends on Csm4. In Mps2-depleted cells ($P_{CLB2}$-MPS2), Csm4, but not Mps3, was no longer detectable at the nuclear periphery and became diffused throughout the cell (Fig 2E). Alternatively, the lack of anchoring at the nuclear periphery may result in a rapid degradation of Csm4. In addition, the meiotic nucleus became spherical without any visible membrane protrusions (Fig 2E and F). As in csm4Δ cells, Mps3 was distributed evenly around the nuclear periphery when Mps2 was absent (Fig 2E, right panels). Using the perfect sphere as a reference, we determined nuclear roundness at prophase I (Fig 2F). Both Mps2 and Csm4 were required for nuclear deformation; importantly, no additive effect was observed in the csm4Δ $P_{CLB2}$-MPS2 double mutant (Fig 2F), indicating Mps2 and Csm4 act in the same pathway. Finally, we found that localization of Mps2 to the nuclear periphery, but not to the SPB, was impaired when Mps3 was depleted in yeast meiosis (Fig S2A), indicating that Mps3 regulates the association of Mps2 with the nuclear envelope. Consequently, nuclear protrusions were not observed in cells depleted with meiotic Mps3 (Fig S2B). Taken together, our findings demonstrate that localization of Csm4 to the nuclear periphery depends on Mps2, but not vice versa. Furthermore, Mps2 and Csm4 both are essential for generating nuclear protrusions, a major t-LINC complex activity, at prophase I.

### Meiotic Mps2 is a telomere-associated protein

If Mps2 is a component of the t-LINC, we reasoned that Mps2 would localize to the telomere, as Csm4 and Mps3 do (Conrad et al, 2007, 2008). We, therefore, performed surface nuclear spreads, in which telomere-associated proteins can be determined by immunofluorescence (Fig 3). As a reference, cohesin Rec8 was used as a marker for the meiotic chromosome axis (Fig 3A–D). We found that Mps2 colocalized with Ndj1 at the telomeres, with each Ndj1 focus associated with a corresponding Mps2 focus in the cell shown in Fig 3A. In addition, both Mps2 and Csm4 were colocalized at the chromosome ends (Fig 3B). Therefore, meiotic Mps2 is a telomere-associated protein.

We set out to determine the factors that regulate Mps2 binding to the telomere. We found that in the absence of Csm4, Mps2 remained bound to the chromosome ends, indicating that Csm4, which is also localized to the ONM, is not required for Mps2's association with the telomere (Fig 3C). This finding is consistent with the observation that nuclear localization of Mps2 is independent of Csm4 (Fig 2D). In contrast, removal of Mps3 or Ndj1 abolished Mps2's binding to the chromosome ends (Fig 3C). Therefore, Mps3 and Ndj1 are required for telomere localization of Mps2. These findings prompted us to

hypothesize that Mps2 acts as a linker between Mps3 and Csm4 to mediate t-LINC complex formation. Indeed, depletion of meiotic Mps2 abolished Csm4's localization to the telomere (Fig 3D). In contrast, depletion of Mps2 did not alter Mps3's association with the telomere (Fig S2C). Taken together, our findings indicate that the yeast t-LINC is composed of Csm4, Mps2, and Mps3 and that Mps2 links Csm4 and Mps3 together.

### Mps2 regulates telomere bouquet formation and meiotic recombination

At prophase I, the t-LINC complex mediates telomere bouquet formation and regulates meiotic recombination (Conrad et al, 2008; Kosaka et al, 2008; Koszul et al, 2008; Wanat et al, 2008). To determine the role of Mps2 in meiotic recombination, first, we asked whether Mps2 is required for telomere bouquet formation. We used the telomere-associated protein Rap1 (Conrad et al, 1990), which is tagged with GFP, to serve as a telomere marker. Rap1-GFP formed distinctive foci at the nuclear periphery at prophase I (Fig 4A). In wild-type cells, Rap1 foci were clustered together and often occupied half or less than half of the nuclear periphery, revealing the telomere bouquet configuration at prophase I (Fig 4A and B). In contrast, depletion of meiotic Mps2 or removal of Csm4, or both, abolished the telomere bouquet formation (Fig 4A and B). Of note, the bouquet configuration appeared to be transient at prophase I. Whereas telomere bouquet formation took place ubiquitously in individual cells, it was observed in about 25% of the cells in a population 3 h after the induction of meiosis (Fig 4B). We concluded that like Csm4, Mps2 is required for telomere bouquet formation.

Next, we determined homolog pairing with the TetO/TetR-GFP system that marks chromosome IV at the LYS4 locus (Fig S2D and Jin et al, 2009). In the absence of Mps2, meiotic cells appeared competent to pair at the LYS4 locus, but displayed a 2-h delay in pairing (Fig S2D). This was also the case in csm4Δ cells (Fig S2D). Finally, we observed that the gene conversion rate at the ARG4 and HIS4 loci reduced about 10-fold in both $P_{CLB2}$-MPS2 and csm4Δ cells compared with those of the wild type (Fig 4C–E). Together, these findings suggest that Mps2 is required for efficient homolog pairing and meiotic recombination and further support the notion that Mps2 is a component of the t-LINC complex. The delayed homolog pairing and reduced gene conversion rate observed in $P_{CLB2}$-MPS2 cells also indicate that as a component of the t-LINC complex, Mps2 may play a role in resolving chromosome interlocks during early meiosis.

### Reconstitution of t-LINC in vegetative yeast cells

Among the three components of the t-LINC complex, only Csm4 is specific to meiosis. We therefore hypothesized that by ectopically expressing CSM4, the t-LINC complex would be reconstituted in vegetative yeast cells. We used $P_{GAL1}$-CSM4 to induce Csm4 production in cells grown in galactose medium (Fig 5). Ectopic expression of CSM4 caused a slow growth defect (Fig 5A). Crucially, this mutant phenotype was suppressed by the overproduction of Mps2 by way of $P_{GAL1}$-MPS2 (Fig 5A), demonstrating that CSM4 and MPS2 genetically interact. In the presence of Csm4, we observed that an ectopic patch of Mps3, but not the SPB marker Tub4, formed in the developing daughter cell during mitosis (Figs 5B–D and 6A). Cells

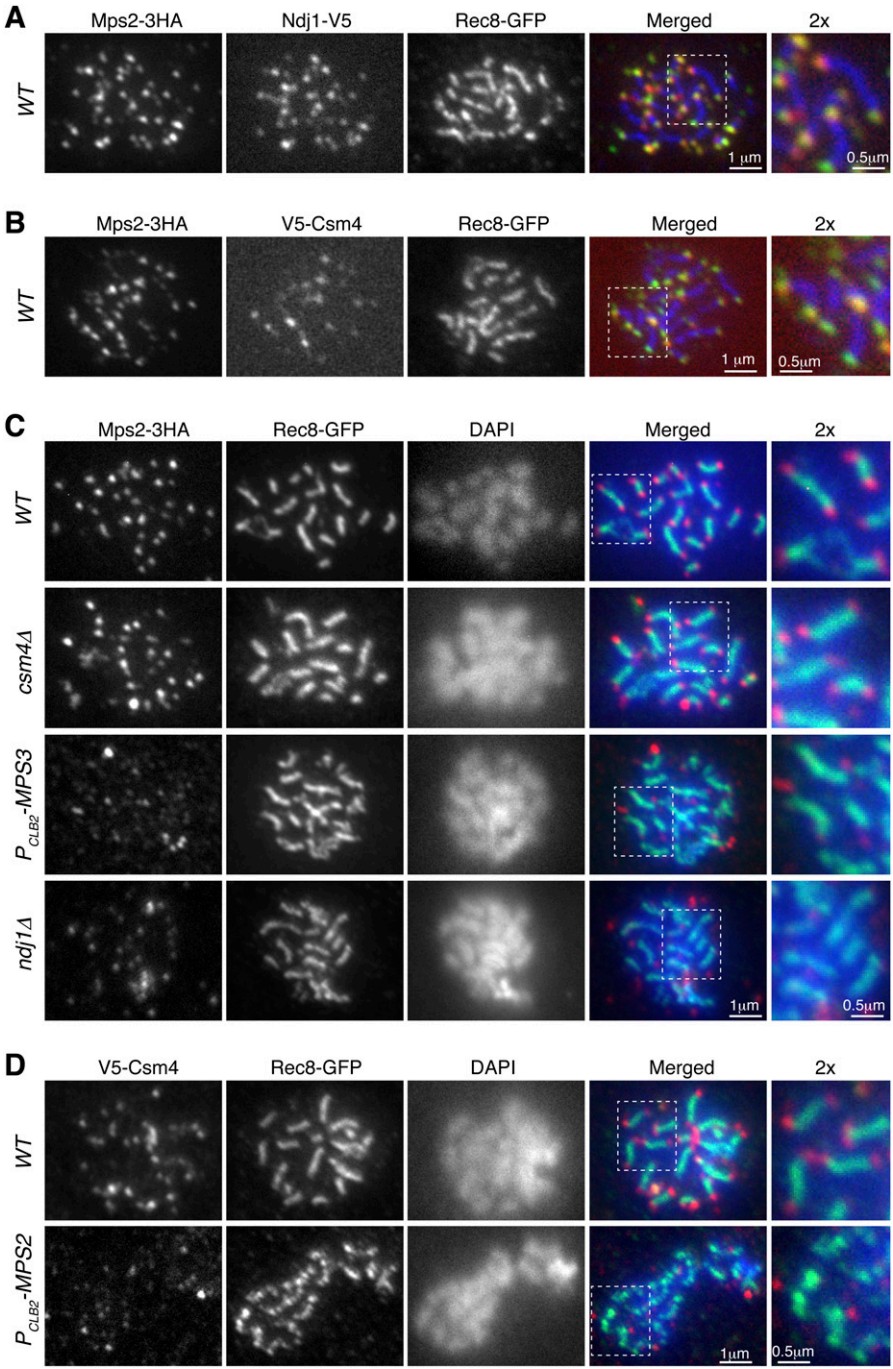

**Figure 3. Mps2 is a telomere-associated protein.**
Meiotic cells were harvested for nuclear spreads, followed by immunofluorescence to probe V5-, HA-, and GFP-tagged proteins. DAPI stains DNA. Rec8 is used to mark the chromosome axis. Enlarged views (2×) are shown to the right. **(A)** Representative cell showing colocalization of Mps2 with Ndj1 to meiotic telomeres. Red, Ndj1-V5; green, Mps2-3HA; blue, Rec8-GFP. **(B)** Representative cell showing colocalization of Mps2 and Csm4 at telomeres. Red, V5-Csm4; green, Mps2-3HA; blue, Rec8-GFP. **(C)** Representative cells showing that telomeric localization of Mps2 depends on Mps3 and Ndj1 but not Csm4. Red, Mps2-3HA; green, Rec8-GFP; blue, DAPI. Note that Mps2 localizes to the chromosome ends in *WT* and *csm4Δ* cells. **(D)** Representative cells showing telomeric localization of Csm4 depends on Mps2. Note that chromosome axes appear less compacted in the *P$_{CLB2}$-MPS2* cell. Red, V5-Csm4; green, Rec8-GFP; blue, DAPI.

with ectopic Csm4 showed a delayed mitotic program (Fig 5E), consistent with the slow growth phenotype of *P$_{GAL1}$-CSM4* (Fig 5A). Formation of this Mps3 patch corresponded to the precocious extension of the nuclear envelope into the daughter cell before chromosome segregation in mitosis (Fig S3A). Crucially, both Mps2 and Mps3 were located at the leading edge of this Csm4-dependent nuclear extension (Fig S3B), which is reminiscent of the nuclear protrusions mediated by the t-LINC complex at meiotic prophase I (Fig 2D and E).

To determine whether the ectopic Mps3 patch depends on Mps2, forming an intact t-LINC complex, we created an *mps2Δ* strain (Fig S4). Because *MPS2* is an essential gene, we took advantage of the fact that *pom152Δ* suppresses the lethal phenotype of *mps2Δ*, presumably bypassing the need of Mps2 in SPB duplication (Fig S4A and Katta et al, 2015). In *mps2Δ pom152Δ* double-mutant cells, we never observed the ectopic Mps3 patch in the daughter cell with or without the presence of Csm4 (Fig S4B). Together, these findings demonstrate that both Csm4 and Mps2 are required for the

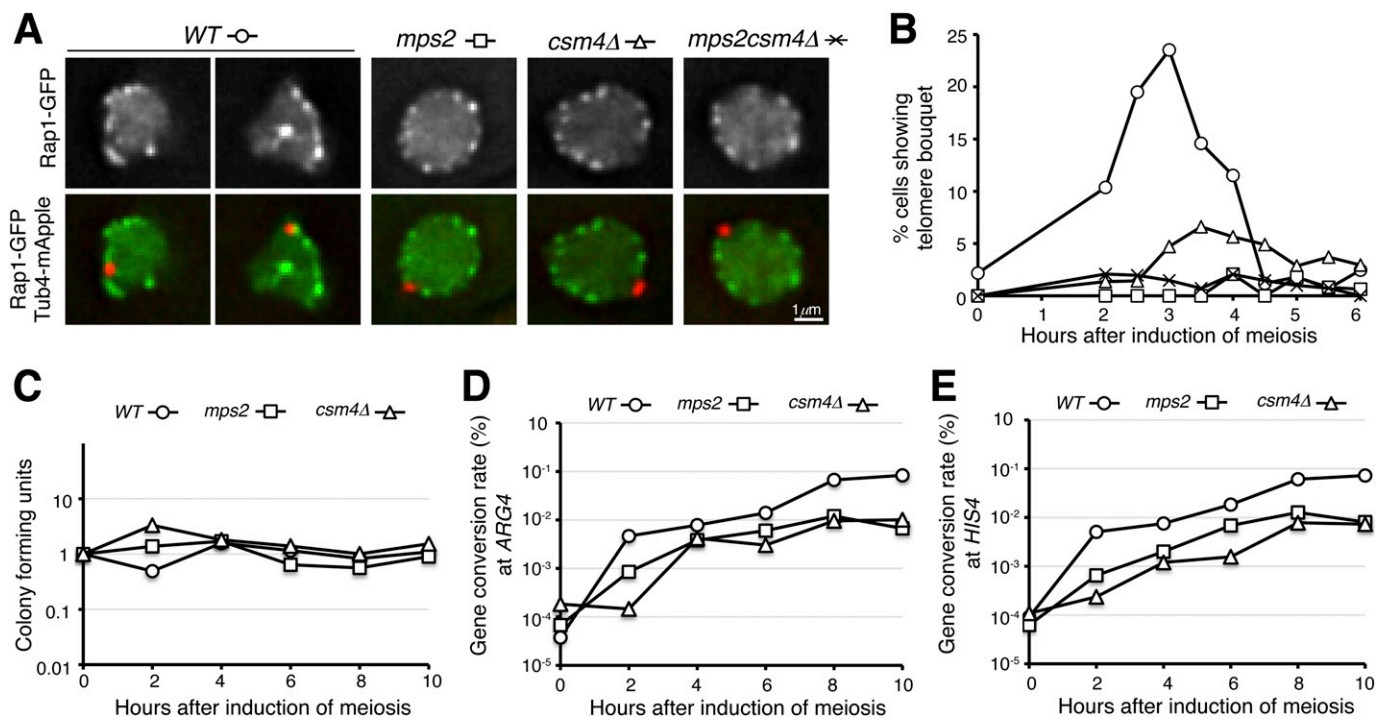

**Figure 4. Mps2 regulates telomere bouquet formation.**
**(A)** Representative images showing Rap1-GFP distribution in *WT*, $P_{CLB2}$-*MPS2*, *csm4Δ*, and $P_{CLB2}$-*MPS2 csm4Δ* cells at prophase I. Red, Tub4-mApple; green, Rap1-GFP.
**(B)** Quantification of telomere bouquet formation in *WT*, $P_{CLB2}$-*MPS2*, *csm4Δ*, and $P_{CLB2}$-*MPS2 csm4Δ* cells. Three biological replicates were performed; one representative is shown. At least 100 cells were counted at each time point. **(C, D, E)** Gene conversion rate at the *ARG4* and *HIS4* loci in *WT*, $P_{CLB2}$-*MPS2*, and *csm4Δ* cells. Yeast cells were induced to undergo synchronous meiosis; aliquots were withdrawn at indicated times. Serially diluted yeast cells were plated on yeast extract, peptone and dextrose (YPD) to determine cell viability (panel C) and on selective dropout medium to determine the rates of gene conversion (panels D and E). Three biological replicates were performed; one representative is shown.

formation of the ectopic Mps3 patch, and therefore the intact t-LINC complex, in the daughter cell during mitosis.

Because the t-LINC complex mediates actin-based motility in meiosis (Koszul et al, 2008), we reasoned that the formation of the ectopic Mps3 patch and thereby the t-LINC complex depends on actin polymerization, which is highly active in the budding daughter cell (Bi & Park, 2012). To test this hypothesis, we treated yeast cells with the actin polymerization inhibitor latrunculin B (Lat B) (Fig 5F). Mps3 patches disappeared 15 min after the addition of Lat B to the yeast medium (Fig 5F), suggesting that formation of the ectopic t-LINC complex in the daughter cell depends on actin polymerization. Together, these findings demonstrate that the t-LINC complex can be reconstituted in vegetative yeast cells simply by ectopic production of Csm4 and that Mps2 acts as the linker that connects Csm4 to Mps3, confirming the heterotrimeric nature of the t-LINC complex in budding yeast.

### Reconstituted t-LINC complex can tether telomeres

To determine whether reconstituted t-LINC complex is capable of tethering telomeres, we induced the production of Ndj1 together with Csm4 in vegetative yeast cells (Fig 6B). We have shown previously that Ndj1, when ectopically produced in mitosis, binds to Mps3 (Li et al, 2015). In the presence of Ndj1, we predicted that the t-LINC complex tethers telomeres to the nuclear envelope. In wild-

type cells, telomeres, marked by Rap1-GFP, trailed the separating SPBs during mitosis (Fig 6B). In contrast, in the presence of both Csm4 and Ndj1, Rap1-GFP entered the daughter cell precociously, forming an ectopic patch just like the Mps3 patch, well before SPB separation (Fig 6B). Therefore, ectopically reconstituted t-LINC complex is functional in tethering telomeres.

## Discussion

We have demonstrated the heterotrimeric composition of the budding yeast t-LINC complex; specifically, the KASH-like protein Mps2 bridges Csm4 and Mps3 (Fig 7). Four lines of evidence support the idea of a heterotrimeric nature of the yeast t-LINC complex. First, Mps2 is a major binding partner of Csm4 and colocalizes with Csm4 at the telomere. Second, Mps2 is required for Csm4's association with the telomere, but not for Mps3. Third, Mps2 is essential for telomere bouquet formation, a major activity of the t-LINC complex. Finally, Mps2 is required for functional reconstitution of the t-LINC complex in vegetative yeast cells by linking Csm4 and Mps3 together. We note that a recent work also suggests that Mps2 acts as a member of the t-LINC complex (Lee et al, 2020). In budding yeast, the combined action of Mps2 and Csm4 is needed to carry out the function of the canonical KASH protein at the telomere. Mps2 is thought to form a homo-oligomer through its coiled-coil regions

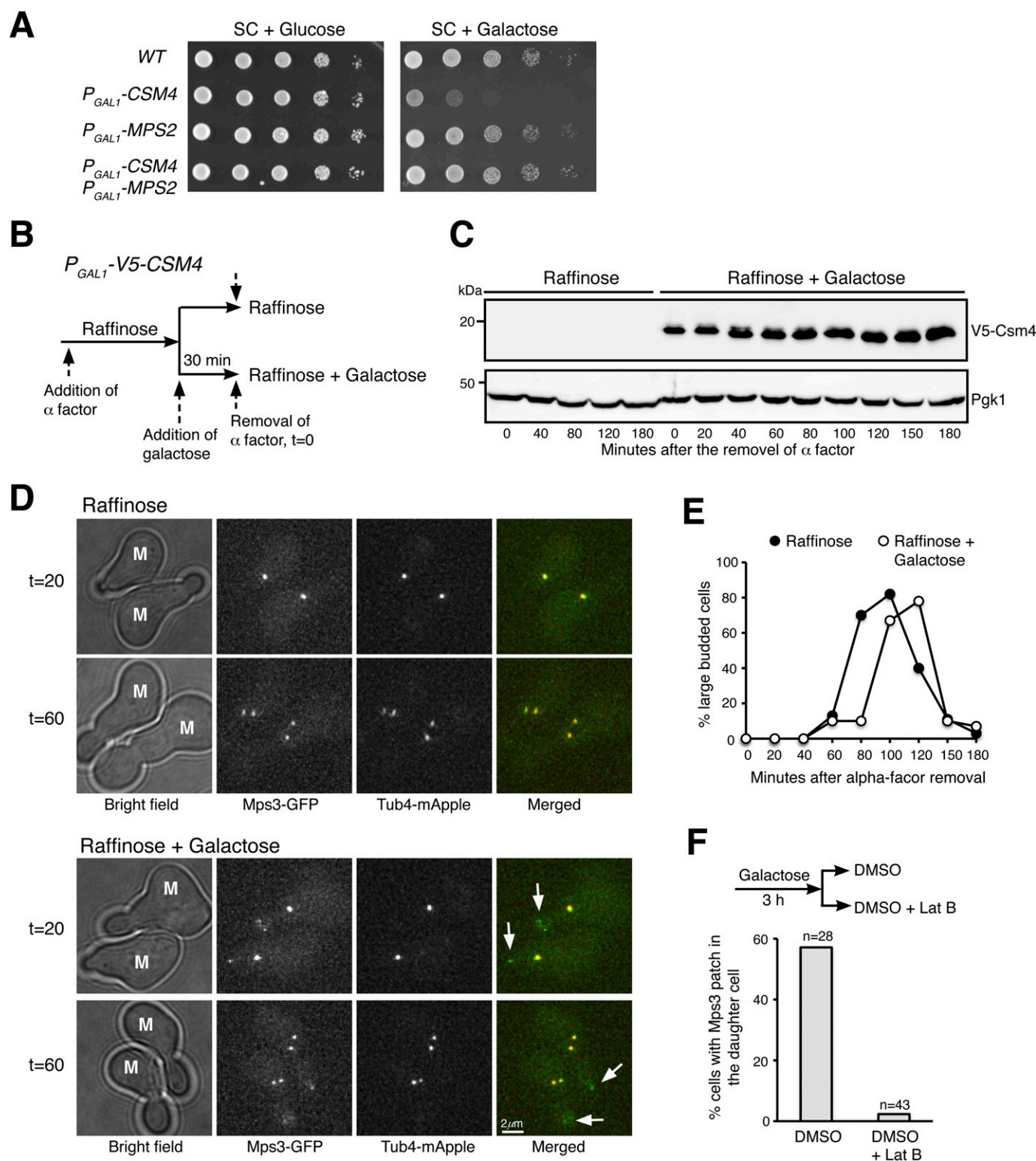

**Figure 5. Ectopic production of Csm4 reconstitutes t-LINC complex in mitosis.**
**(A)** Genetic interaction between *MPS2* and *CSM4*. 10-fold diluted yeast cells were spotted onto glucose or galactose medium. Note that ectopic expression of *CSM4* in the galactose medium is toxic to the vegetative yeast cell. **(B)** Schematic diagram showing the experimental procedure for panels (C, D, E). **(C)** Western blotting showing induced production of Csm4 in the galactose medium. V5-Csm4 was probed by an anti-V5 antibody. The level of Pgk1 serves as a loading control. **(D)** Formation of the Mps3 patch in the daughter cell in the presence of Csm4. Tub4-mApple (red) marks the spindle pole body. Projected images of 12 z-sections are shown. Arrows point to the Mps3-GFP (green) patch in daughter cells. **(E)** Quantification of budding index. Cell aliquots were withdrawn at indicated times, and budding morphology was determined by phase-contrast microscopy. More than 200 cells were counted at each time point in both raffinose and galactose treatments. **(F)** Impact of actin polymerization on Mps3 patch formation. Schematic diagram at the top shows the experimental procedure. Fluorescence microscopy was performed 15 min after the treatments to visualize Mps3-GFP patch formation as in panel (C). Lat B, latrunculin B. M, mother cell; SC, synthetic complete.

(Zizlsperger & Keating, 2010); importantly, sequence similarity between Csm4 and Mps2 is confined to their coiled-coil regions (Fig 1A). In addition, Bbp1, which competes with Csm4 (our unpublished data), binds to Mps2 at the coiled coils (Kupke et al, 2017). We therefore speculate that the yeast t-LINC complex is a nonamer (Fig 7). In this model, we postulate that Csm4 directly interacts with the cytoskeleton to link telomeres to the actin filament and its associated motor proteins. Alternatively, Csm4 could act as a regulator that modulates Mps2's binding affinity to the actin-based motor protein Myo2 (Lee et al, 2020). Because Mps2 and Mps3 are members of both the c-LINC and t-LINC complexes, we speculate a crosstalk takes place between them to ensure that centrosome dynamics is coordinated with telomere movement in budding yeast meiosis.

In conclusion, ONM-localized Mps2 and Csm4 act together to function as a canonical KASH protein at the t-LINC complex in budding yeast. This action is analogous to that of the WIP and WIT proteins in *Arabidopsis* (Meier, 2016) and that of Klar and Msp-300 in *Drosophila* (Elhanany-Tamir et al, 2012). In metazoans, numerous KASH variants also exist. Our work suggests that variant LINC complexes could be prevalent and provides insight into LINC complex assembly and its evolution in eukaryotes.

# Materials and Methods

### Yeast strains and plasmids used in this study

Yeast strains and plasmids used in this study are listed in Tables S1 and S2. Strains for meiotic experiments are isogenic to the SK1 genetic background and strains for mitotic experiments are from the S288C background. To generate proteins that are tagged at their N termini, alleles of *TAP-MPS2*, *V5-CSM4*, *TAP-CSM4*, *V5-MPS2*, *TAP-MPS3*, *GFP-MPS2*, and *GFP-CSM4* were created by homologous recombination-based gene replacement as we have described previously (Koch et al, 2019). Briefly, the corresponding plasmids (Table S2) were linearized by restriction digestion and integrated at the endogenous locus of each respective allele by yeast transformation. To remove the untagged gene, *URA3*-positive colonies were then counter-selected on a 5-fluoroorotic acid (5-FOA) plate; these tagged alleles therefore served as the only functional copy in the yeast genome. All of these alleles were functional and verified by DNA sequencing before use.

To generate C-terminal–tagged alleles, a PCR-based yeast transformation method (Longtine et al, 1998) was used to generate *CSM4-GFP*, *MPS2-RFP*, and *MPS2-3HA*. Positive transformants were confirmed by colony PCR. A comparable PCR-based method was used to replace the *CSM4*, *MPS2*, and *POM152* open reading frames with either a KanMX4 or hygromycin-B cassette to generate gene deletions. Correct transformations were further confirmed by colony-based diagnostic PCR. Using a similar PCR-based method, $P_{CLB2}$-*MPS2* was generated by replacing the endogenous promoter with the mitosis-specific promoter from *CLB2* (Lee & Amon, 2003). Primers used in this study are included in Table S3. The following alleles have been reported previously: *MPS3-V5*, *ndt80Δ*, *MPS3-mApple*, *TUB4-mApple*, *NDJ1-V5*, *HTA1-mApple*, *REC8-GFP*, $P_{CLB2}$-*MPS3*, *ndj1Δ*, *RAP1-GFP*, and *MPS3-GFP* (Li et al, 2014, 2015).

To ectopically express *CSM4* in vegetative yeast cells, we constructed *PGAL1-V5-CSM4* (pHG317) to express the full-length *CSM4*

under the control of the *GAL1* promoter. We linearized plasmid pHG317 with PstI and integrated it at the endogenous *CSM4* locus by yeast transformation. A similar approach was used to overexpress *MPS2* in vegetative yeast cells by constructing *PGAL1-GFP-MPS2* (pHG527). Plasmid pHG527 was linearized with StuI and integrated at the endogenous *MPS2* locus by yeast transformation. Note that the endogenous *MPS2* remains intact and functional. The plasmid pHG335 (*PGAL1-V5-NDJ1*) has been described previously (Li et al, 2015). Leucine-positive colonies were confirmed by colony-based diagnostic PCR.

### Yeast culture method and cell viability assay

For meiotic experiments, yeast cells were grown in YPD (1% Yeast extract, 2% Peptone, and 2% Dextrose) at 30°C. These YPD cultures were diluted with YPA (1% Yeast extract, 2% Peptone, and 2% Potassium Acetate) to reach OD (optical density, λ = 600 nm) of 0.2 and incubated at 30°C for ~14 h to reach a final OD of ~1.6–1.8. Yeast cells were then washed by water and resuspended in 2% potassium acetate to induce synchronous meiosis as described previously (Li et al, 2015). Yeast samples were withdrawn at the indicated times for fluorescence microscopy and/or protein extraction.

To synchronize cycling cells, S288C yeast cells were grown in SC (synthetic complete) medium with 2% raffinose to an OD of 0.5 and arrested at the G1 phase by the addition of 10 μg/ml (final concentration) of α factor. Raffinose cultures were separated into two flasks after the addition of α factor; one served as the control and the other one received galactose (2% final concentration) 30 min before α factor removal. The addition of galactose induced the expression of *GAL*-regulated genes before the release of yeast cells from G1 arrest. To remove the α factor, the cells were washed twice with water and once with SC raffinose or galactose, and then resuspended in the respective medium. Samples were withdrawn at the indicated times for fluorescence microscopy and/or protein extraction (Koch et al, 2019).

To determine cell growth, yeast cells were grown overnight to reach saturation in YPD liquid medium, 10-fold diluted, spotted onto SC plates containing either 2% glucose or 2% galactose and then incubated at 30°C for about 2 d.

### Protein-affinity purification and mass spectrometry

Protein-affinity purification was performed as we have reported previously (Li et al, 2015). In brief, 2 liters of yeast cells were induced into synchronous meiosis for 6 h. Yeast cells were harvested and ground into powder in the presence of liquid nitrogen. The yeast powder was then stored at –80°C before use. For affinity purification, yeast powder was thawed in the extraction buffer. The lysate was then incubated with epoxy-activated M-270 Dynabeads (Cat. no. 14305D; Thermo Fisher Scientific), which were cross-linked with rabbit IgG (Cat. no. I5006; Sigma-Aldrich). The final product was eluted from the beads and dried for further study.

Purified protein samples were digested by trypsin. The proteomics work was carried out by the Translational Science Laboratory, Florida State University College of Medicine. An externally calibrated Thermo

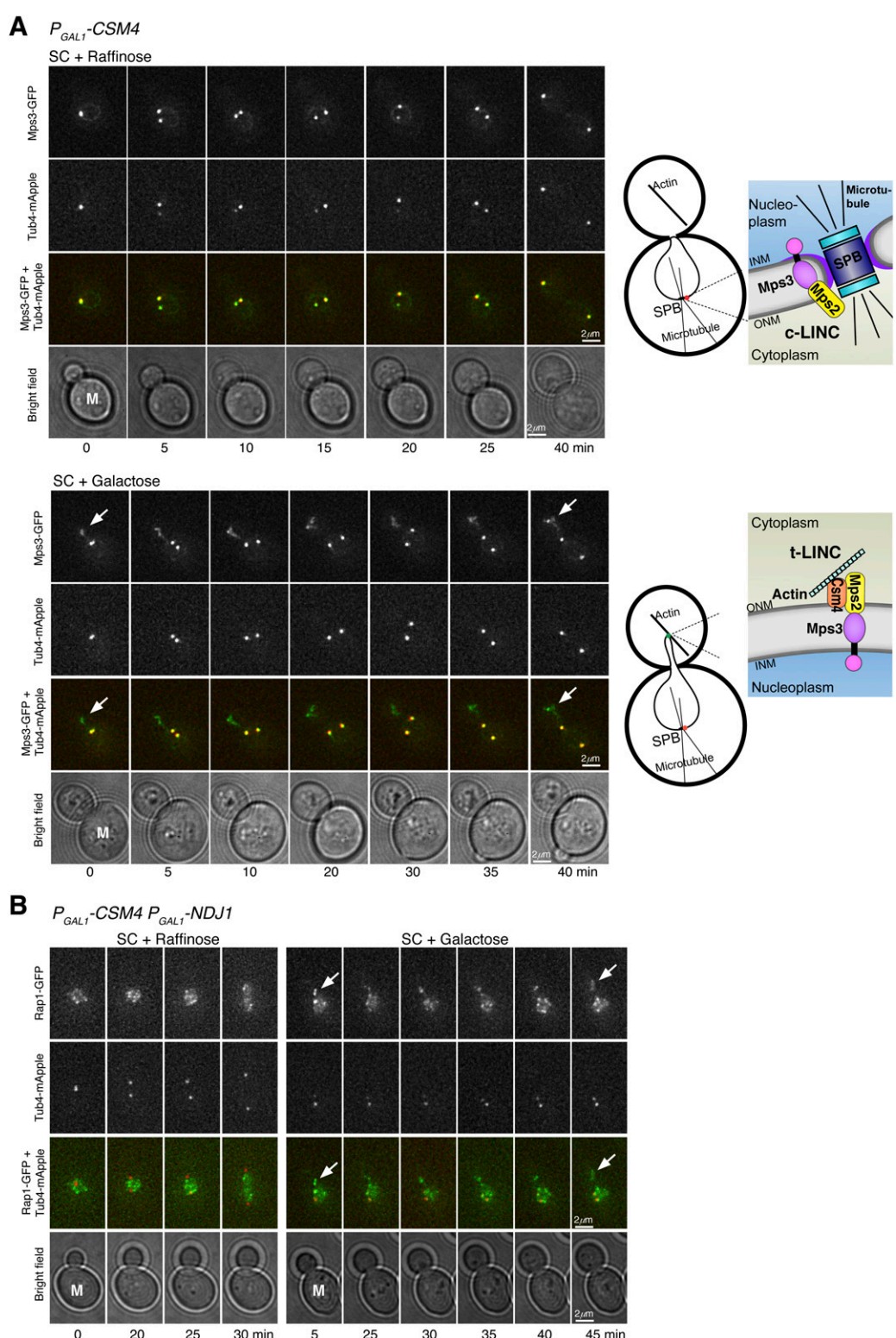

**Figure 6. Reconstitution of t-LINC complex in vegetative yeast cells.**
**(A)** Induction of t-LINC complex formation in mitosis with ectopic Csm4. Time-lapse fluorescence microscopy showing the localization of Mps3-GFP (green). Tub4-mApple (red) marks the spindle pole body (SPB). Arrows pointing to the Mps3 patch formed in the developing daughter cell when Csm4 was produced. Projected images from 12 z-sections are shown. Time zero refers to the onset of SPB separation. Schematic diagrams of c-LINC and t-LINC complexes are shown to the right. **(B)** Reconstituted t-LINC complex tethers telomeres. Time-lapse fluorescence microscopy was performed as above. Rap1-GFP (green) marks the telomeres; Tub4-mApple (red) marks the SPB. Projected images from 12 z-sections are shown. Time zero refers to the onset of SPB separation. Note that in the presence of Csm4, Rap1-GFP formed a patch in the developing daughter cell. INM, inner nuclear membrane; M, mother cell; ONM, outer nuclear membrane.

LTQ Orbitrap Velos mass spectrometer was used for mass spectrometry as per the method described previously (Li et al, 2015).

## Protein extraction and Western blotting

For meiotic yeast cells, proteins were extracted with the trichloroacetic acid (TCA) method as described previously (Jin et al, 2009). In brief, 3–5 ml of yeast cells was collected, resuspended in 2.5% ice-cold TCA, and incubated at 4°C for 10 min. Cell pellets were stored at −80°C before use, and proteins were extracted in the RIPA buffer by bead beating with a mini bead-beater homogenizer for 90 s at 4°C before standard SDS–PAGE and Western blotting.

For mitotic experiments, yeast aliquots were withdrawn at the indicated times for protein extraction by precipitation in the presence of 20 mM NaOH and standard SDS–PAGE and Western blotting protocols were followed (Koch et al, 2019).

Proteins tagged with HA (Mps2-3HA and 3HA-Mps2) were detected by an anti-HA mouse monoclonal antibody (1:1,000 dilution, 12CA5; Sigma-Aldrich). Similarly, V5-tagged proteins (V5-Csm4, Mps2-V5, and Mps3-V5) were detected by an anti-V5 mouse monoclonal antibody (1:1,000 dilution, Cat. no. 66007-1-Ig; Proteintech), and TAP-tagged proteins (TAP-Csm4, TAP-Mps2, and TAP-Mps3) were detected by an anti-TAP rabbit antibody (1:10,000, Cat. no. CAB1001; Thermo Fisher Scientific). The level of Pgk1 was detected by a Pgk1 antibody (1:10,000, Cat. no. PA5-28612; Thermo Fisher Scientific) and was used as a loading control. Horseradish peroxidase–conjugated secondary antibodies, goat antimouse, and goat antirabbit (Cat. no. 1706516 and 1705046; Bio-Rad) were used to probe the proteins of interest by an ECL kit (Cat. no. 1705060; Bio-Rad). Two ECL-based Western blot detection methods were used, X-ray film (Figs 1–5) and the ChemiDoc MP Imaging System (Cat. no. 17001402; Bio-Rad) (Fig S3).

## Live-cell fluorescence microscopy

Live-cell fluorescence microscopy was conducted on a DeltaVision imaging system (GE Healthcare Life Sciences) with a 63× objective lens (NA = 1.40) on an inverted microscope (IX-71; Olympus) and with xenon arc lamp illumination. Microscopic images were acquired with a CoolSNAP HQ2 CCD camera (Photometrics). Before microscopy, yeast cells were prepared as described previously (Li et al, 2015). Briefly, the yeast cells were prepared on a concave microscope slide (~0.8 mm deep) filled with an agarose pad with 2% potassium acetate. The concave slide was then sealed with a cover slip and scoped for the desired time duration. The microscope stage was enclosed in an environmental chamber set at 30°C. For time-lapse microscopy, optical sections were set at 0.5-$\mu$m-thickness with 12 z-sections. Ultrahigh signal-to-background coated custom filter sets were used. For GFP, the excitation spectrum was at 470/40 nm, emission spectrum at 525/50 nm; for RFP, excitation was at 572/35 nm and emission at 632/60 nm. To minimize photo toxicity to the cells and photo bleaching to fluorophores, we used neutral density filters to limit excitation light to 32% or less of the normal equipment output for time-lapse microscopy. Images were deconvolved with SoftWoRx (GE Healthcare Life Sciences); projected images or single optical sections were used for display.

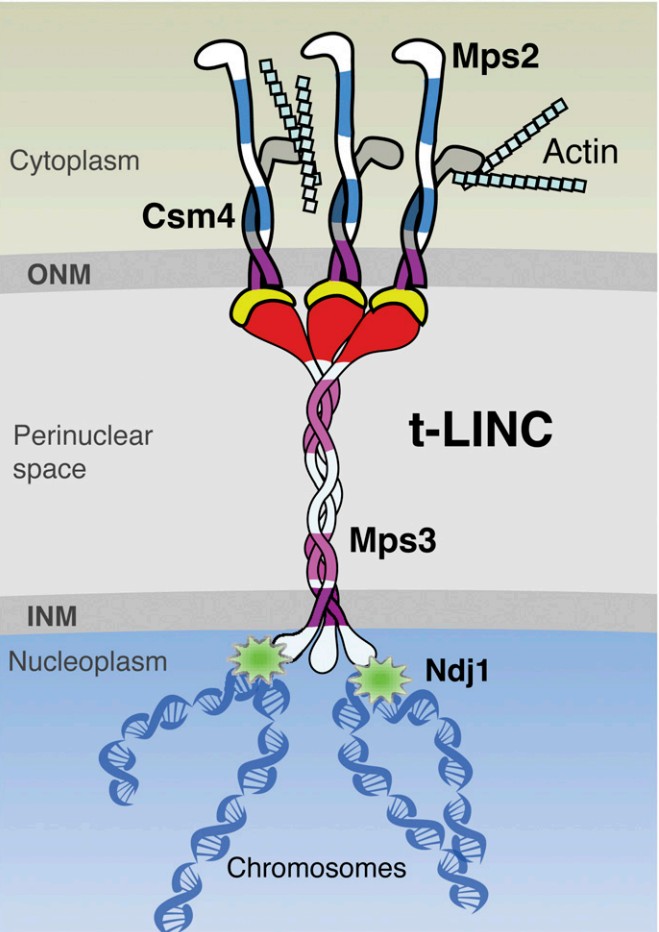

**Figure 7. Model for t-LINC complex in budding yeast.**
Three copies of each of Csm4, Mps2, and Mps3 are proposed to form a t-LINC nonamer. INM, inner nuclear membrane; ONM, outer nuclear membrane.

To determine meiotic cell progression, aliquots of yeast cells were collected at indicated times and prepared for fluorescence microscopy. Tub4-mApple serves as an SPB marker. At least 100 cells were counted at each time point to determine the rate of SPB separation.

In experiments testing the dependence of actin filaments, latrunculin B (final concentration of 100 $\mu$M) (Koszul et al, 2008) was added to the cell culture before microscopy. The same volume of DMSO was added in the control group.

To determine the nuclear roundness factor in cells staged at prophase I, the median section of the nucleus from the z-stacks was measured by freehand tracing in ImageJ. The formula 4 × area/($\pi$ × major_axis^2) is used to calculate the roundness factor.

## Nuclear spread and immunofluorescence

Surface nuclear spreads were performed as described previously (Jin et al, 2009). In brief, yeast cells enriched at prophase I (~5 h after induction of meiosis) were spheroplasted by lyticase treatment. Spheroplasts were then fixed and poured onto a glass slide. The slide was then rinsed with PhotoFlo 200 and air-dried, followed by PBS buffer with 3% BSA to block for 2 h at room temperature. Anti-V5 antibody (R960-25; Thermo Fisher Scientific) was used to detect

V5-Csm4 and Ndj1-V5; anti-HA antibody (12CA5; Roche/Sigma-Aldrich) was used to detect Mps2-3HA. Rec8-GFP was detected by an anti-GFP mouse monoclonal antibody (ab209; Abcam). Secondary antibodies (FITC-conjugated goat antirabbit, rhodamine-conjugated goat anti-mouse, and Cy3-conjugated goat antirat; Jackson ImmunoResearch Laboratories) were used at a dilution of 1:500. Mounting medium with DAPI was added before microscopy. Images were acquired with an epifluorescence microscope (Axio Imager M1; Zeiss) with a 100× objective lens (NA = 1.40) at room temperature.

### Gene conversion assay

Yeast cells were induced to undergo synchronous meiosis, and aliquots were withdrawn at the indicated times. Serially diluted yeast cells were plated on YPD plates to determine cell viability and on SC arginine-dropout and SC histidine-dropout plates to determine gene conversion rate at the *ARG4* and *HIS4* loci. The rate of gene conversion was calculated by the ratio of the colony-forming units on SC dropout plates over those on YPD plates.

## Supplementary Information

## Acknowledgements

We thank Yanchang Wang and Robert Tomko for insightful discussions. Elizabeth Staley provided technical support. Jen Kennedy and Charles Badland assisted with text editing and graphic design, respectively. This work was supported by the National Institute of General Medicine (GM117102,138838) and the National Science Foundation (MCB1951313).

### Author Contributions

J Fan: data curation, formal analysis, validation, investigation, visualization, methodology, and writing—review and editing.
H Jin: data curation, formal analysis, validation, investigation, methodology, and writing—review and editing.
BA Koch: data curation, formal analysis, validation, investigation, visualization, and writing—review and editing.
H-G Yu: conceptualization, data curation, formal analysis, supervision, funding acquisition, validation, investigation, visualization, methodology, project administration, and writing—original draft, review, and editing.

### Conflict of Interest Statement

The authors declare that they have no conflict of interest.

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
