## [Reviewer comments · Life Science Alliance]

Life Science Alliance

Mps2 links Csm4 and Mps3 to form a telomere-associated LINC complex in budding yeast

Jinbo Fan, Hui Jin, Bailey Koch, and Hong-Guo Yu

DOI: <https://doi.org/10.26508/lsa.202000824>

Corresponding author(s): Hong-Guo Yu, Florida State University

Review Timeline:

Submission Date:	2020-06-22
Editorial Decision:	2020-08-10
Revision Received:	2020-08-31
Editorial Decision:	2020-09-10
Revision Received:	2020-09-10
Accepted:	2020-09-11

Scientific Editor: Shachi Bhatt

Transaction Report:

August 10, 2020

Re: Life Science Alliance manuscript #LSA-2020-00824-T

Dr. Hong-Guo Yu
Florida State University
Biological Science
89 Chieftan Way
Tallahassee, Florida 32306

Dear Dr. Yu,

Thank you for submitting your manuscript entitled "Mps2 links Csm4 and Mps3 to form a telomere-associated LINC complex in budding yeast" to Life Science Alliance. We apologize for this delay in getting our decision back to you.

The manuscript was assessed by expert reviewers (reports attached at the end of this email), and as you can see, all three reviewers have shown enthusiasm for the findings in this study, but have also pointed out some concerns that need to be addressed prior to publication. We encourage you to re-submit a revised version of the manuscript that addresses all of the reviewers' concerns. While we appreciate the detailed review by Referee # 2, figuring out what happens to the GFP-Csm4 construct would not be required experimentally; experimental evidence for this point (R2 pt 7c) should only be included in the revised manuscript, if readily attainable, otherwise a discussion should suffice.

Thank you for this interesting contribution to Life Science Alliance. We are looking forward to receiving your revised manuscript.

Sincerely,

Reilly Lorenz
Editorial Office Life Science Alliance
Meyerhofstr. 1
69117 Heidelberg, Germany
t +49 6221 8891 414
e contact@life-science-alliance.org
www.life-science-alliance.org

B. MANUSCRIPT ORGANIZATION AND FORMATTING:

Reviewer #1 (Comments to the Authors (Required)):

This is a beautiful paper in conception, execution and presentation. I have only minor comments.

The budding yeast meiotic "t-LINC" complex transduces information across the nuclear envelope (NE), thereby enabling treadmilling of NE-associated cytoplasmic actin fibers to drive telomere-led chromosomal motions within the nucleus. Canonically such complexes include a SUN protein, localizing to the inner nuclear membrane, and a "KASH" protein localized to the outer nuclear membrane, which interact in the peri-nuclear space. The budding yeast has a single SUN-domain protein, Mps3, but lacks a canonical KASH-domain protein. It does have two outer membrane-localized KASH-like proteins, Mps2 and Csm4 (which is meiosis-specific) but exactly how these two molecules might function has been mysterious. However, the authors noted that the structure of Csm4 precludes a KASH-like role in contacting Mps3 through the peri-nuclear space. They therefore sought to identify a missing linker molecule, without bias to possible suspects by asking which molecules co-immunoprecipitate with Csm4. The authors discovered a strong interaction of Csm4 uniquely with Mps2 and confirm the interaction between the two molecules by reciprocal co-IP. They also provide genetic evidence for an interaction via interplay between the two molecules in mitotic cells, where overproduction of (meiosis-specific) Csm4 is deleterious (presumably because it titrates Mps2) and, correspondingly, this deleterious effect is suppressed by overproduction of Mps2.

Extensive additional results show that the combined action of Mps2 and Csm4 is required to carry out the canonical functions of a KASH protein in the t-LINC complex. Specifically, yeast t-LINC is composed of Csm4, Mps2 and Mps3, where Mps2 links Csm4 and Mps3. Another meiosis-specific protein, Ndj1 is required for Mps3 localization to telomeres and thus is required for localization of the entire t-LINC complex.

Thoughtful consideration of all information leads to a specific proposition for the molecular composition of the t-LINC complex and to the suggestion that non-canonical KASH molecules, and thus non-canonical LINC complexes, may be a relatively common occurrence in a variety of situations. Thus, the presented results are likely to be of general significance. The presented results are also interesting and novel with respect to the functional roles of Mps2 (and thus the t-LINC complex).

Additional remarks:

- Csm4 and Mps2 both colocalize to the nuclear periphery during meiosis, as expected from the linker hypothesis. It is also demonstrated for the first time that Mps2 and Mps3 interact. Most importantly, no interaction of Csm4 with Mps3 is detected, in accord with the motivating expectation that such interaction is mediated by a linker molecule (which they now identify as Mps2).
- Mps2 is shown to localize both to the Spindle Pole Body (SPB) and the nuclear periphery, with the specific timing and localization to a subregion of the NE, appropriate to its inferred role. This careful documentation provides a basis for clarity of understanding of dual roles for Mps2 in the two different locations.
- The authors explore the functional significance of Mps2 in a series of rigorous and thoughtful experiments. Since Mps2 has roles also in mitotic cells, they generated and analyzed a pCLB2-MPS2 allele in which meiotic expression is specifically suppressed. Careful cytology directly demonstrates a defect in SPB separation, primarily at MII but also significantly at MI, and appropriate genetic analyses imply that this is a consequence of a direct role of Mps2 at the SPB, rather than any defect in chromosomal processes. The authors do not comment on the fact that exit from MI, even in cells that ultimately execute this division, is delayed and that this delay is eliminated by a spo11 mutation. This is worth noting because it suggests that absence of movement results in recombination-related defects which trigger delayed onset of MI, irrespective of SPB-related failures of MI/II to occur at all.

- Interdependency of t-LINC components for their roles in t-LINC nuclear envelope localization was carefully examined. (a) Mps2 and Mps3 colocalize to a portion of the nuclear periphery. (b) An especially interesting, and elegant finding is that Mps2, Mps3, Csm4 are at leading ends of NE protrusions, which are known to be mediated by telomere-LINC-actin associations. These findings provide direct evidence for localization of these molecules at the appropriate telomere/NE/actin associations. (c) In absence of Csm4, Mps2/Mps3 are still on nuclear periphery but are distributed all around, rather than in a partial domain, and no protrusions occur. Thus, Csm4 is downstream of Mps2/3 for actin-mediated effects. (d) In absence of Mps2, Mps3 still on nuclear periphery but other effects of csm4D absent. Thus, Mps2 is downstream of Mps3. (e) In absence of Mps3, Mps2 localizes aberrantly to the NE (and so there are no nuclear protrusions). (f). Overall: Mps3>Mps2>NE localization; and > Csm4 > NE position bias and protrusions.

- Protrusion data implying colocalization of Mps2 to telomere/NE associations (above) is confirmed by direct analysis of telomeres in spreads. This was very rigorously done using Rec8 as axis marker and Ndj1 as telomere marker. In addition, this assay was used to define functional dependencies for Mps2 telomere association specifically. Telomere association does not require Csm4, the outer membrane protein but does require both Mps3 and Ndj1. Since Ndj1 is required for Mps3 localization to telomeres, which is required for Mps2 localization to telomeres, and in accord with the idea that Mps2 links Mps3 and Csm4, depletion of Mps2 did not affect Mps3 telomere localization but abolished Csm4 localization.

The authors have also carefully defined the functional roles of Mps2 for meiotic chromosomes. (i) They confirm the expectation that Mps2 is required for polarized localization of telomeres to a subdomain of the nuclear envelope (the "bouquet"). They also provide nice evidence that this stage is transient. (ii) They show that close juxtaposition of tagged chromosomal loci ("pairing") occurs as in wild type but that this process is delayed. To this reviewer's eye, this delay comes "late" in prophase, perhaps implying a problem in interlock resolution rather than a problem in pairing per se. This might merit some discussion. (iii) They show that the frequency of heteroallelic recombination, tested at two loci, is reduced 10x. This is very interesting and novel and the explanation is not obvious. It might be due to a lack of pairing; however, there are also more complicated explanations that the authors might wish to consider, particularly in light of the fact that tracts of heteroduplex DNA (which underlie heteroallelic recombination) are longer for interactions that give crossovers than for interactions that give noncrossovers.

The authors next present a lovely set of experiments in which they reconstruct t-LINC activity in vegetatively growing (mitotically dividing) cells. They express the two meiosis-specific components (Ndj1 and Csm4) in vegetative cells. Mps2 and Mps3 are present in such cells as well as in meiosis. They discover that telomeres, marked by Rap1-GFP, enter daughter cells precociously, prior to SPB separation, as seen previously for Mps3. Thus, the t-LINC complex can move telomeres in vegetative cells without addition of any other meiosis-specific components.

The authors draw on published structural information to propose that the yeast t-LINC is a nonamer (Fig 5E). They point clear analogies between the yeast Csm4/Mps2 collaboration with molecules in Arabidopsis and point out that numerous KASH variants also exist in metazoans. They suggest that variant LINC complexes could be prevalent and thus that the current study can provide insight into LINC assembly and its evolution in eukaryotes.

The authors appropriately note that a paper published in April of this year also suggests that Mps2 acts as a component of t-LINC. The two studies have obviously been carried out in parallel. This reviewer finds the current work to be the much more elegant and complete of the two studies and to be a powerful extension to the other study. The current work uses an unbiased approach to identify Mps2 as a relevant component rather than targeting it specifically; provides much more complete characterization of the localization and effects of Mps2 and t-LINC and of Mps2

interactions with other components; provides uniquely provides important functional analysis; and uniquely considers the broader implications of the findings for non-canonical KASH-related proteins. In summary, the current paper is an important, and carefully- and exhaustively-executed contribution to our basic understanding of the budding yeast meiotic t-LINC complex in particular with implications for LINC-mediated complexes in general.

Reviewer #2 (Comments to the Authors (Required)):

This manuscript investigates the assembly and function of LINC complexes in budding yeast. Specifically, it asks how the KASH-like protein Csm4 interacts with the SUN protein Mps3 within the perinuclear space to promote the formation of a telomere-associated LINC (t-LINC) complex. This question is particularly interesting, given the lack of a clear canonical SUN protein-interacting KASH peptide at the C-terminus of Csm4. The authors provide evidence to suggest a model where telomere-associated Mps3 indirectly interacts with Csm4 through Mps2 to form a so-called "heterotrimeric" t-LINC complex, which is required for proper telomere movement and meiotic recombination. In addition, the authors show that expressing the normally meiotic CSM4 in vegetative cells results in the reconstitution of the t-LINC complex that is capable of tethering telomeres to the nuclear envelope. Overall, this intriguing work lays the foundation for understanding the important question of how LINC complexes composed of different KASH proteins are assembled to perform their differential functions in cells. However, before I can recommend this manuscript for publication, the authors need to address the following major and minor issues.

Major Issues

- 1) A major issue that I have with this manuscript is its improper description of the stoichiometry of the LINC complex. For example, the word "heterotrimeric" does not really work to explain the stoichiometry of the t-LINC complex composed of Csm4, Mps2, and Mps3. Based on the literature, what can be said is that Mps3 likely forms a homo-trimer and that a three Mps2 proteins can associate with a Mps3 homo-trimer, resulting in the formation of a hetero-hexameric Mps2-Mps3 LINC complex. Since the stoichiometry of the Csm4-Mps2 interaction is currently unknown, the authors cannot conclude that the yeast t-LINC complex is a nonamer, as they do in the last paragraph of the Results section. Therefore, I would strongly caution against the use of precise stoichiometry to describe t-LINC complexes containing Csm4-Mps2-Mps3.
- 2) It is unclear to me if the fusion proteins used in this work were functional or not. Were these fusions previously characterized? If so, the authors should make this clearer.
- 3) The authors need to better describe the results of the recently published work by Lee et al. (2020). In particular, they need to explain the current thoughts of how the t-LINC complex attaches telomeres to the actin cytoskeleton. By directly addressing the similarities and differences between their work and the work presented in Lee et al., 2020 *Curr Biol*, the authors will help explain their contribution to our understanding of t-LINC complex assembly and function.
- 4) How do the authors envision that Csm4 works together with Mps2-Mps3 to move telomeres? Is it that Csm4 promotes the assembly and function of Mps2-Mps3 LINC complexes? Does Mps2 still interact with Mps3 in cells lacking Csm4? Alternatively, does Csm4 regulate the interaction of Mps2 with Myo2, as suggested by Lee et al., 2020 *Curr Biol*?
- 5) Figures 2C, 4B, 4C, 4D, 4E, S1B, S1C, S1D, S1E, S1F, S1G, and S2D: The authors show plots of

representative experiments and state that they performed 3 biological replicates. I would prefer it if the authors could report a plot of the average measurements from each of the 3 replicates. This would also allow them to perform statistical analyses of these results.

6) Figure 1:

- a. There is a general lack of controls for the TAP experiments presented in this work. Without these controls, it is difficult for me to assess their results.
- b. Why is there a doublet in the Anti-TAP blot for the TAP-CSM4/MPS3-3HA experiment shown in panel C?
- c. It is very hard to see the Mps3-V5 bands in the TAP-MPS2/MPS3-V5 experiment shown in panel D.
- d. How many times were these experiments replicated?
- e. What are the sizes of the z-steps shown in panel E? I am not sure that much is gained from showing these z-sections.
- f. I would like the authors to do the following:
 - i. Perform a structure/function analysis of the Mps2-Csm4 interaction to identify how these proteins associate with each other. Since both proteins have coiled-coil domains, are these required for their interaction?
 - ii. If the authors' hypothesis that Csm4 indirectly interacts with Mps3 via Mps2 were correct, I would anticipate that Mps3 would not be able to immunoprecipitate Csm4 in cells lacking Mps2. Nor would Csm4 be able to immunoprecipitate Mps3 from Mps2-null cells. The authors really need to do these experiments to substantiate their model.
 - iii. Similar to point ii above, images of the nuclear envelope localization of Csm4 in Mps3-null and Mps2-null cells would be helpful.

7) Figure 2:

- a. Quantification of the colocalization of Mps3-mApple with GFP-Mps2 (panel D) or GFP-Csm4 (panel E) would help the authors strengthen their conclusions.
- b. Why does the nuclear envelope look so ratty in the WT cell that express GFP-Csm4 and Mps3-mApple (panel E) compared to nuclear envelope of the WT cell that expresses GFP-Mps2 and Mps3-mApple (panel D)?
- c. What happened to the GFP-Csm4 construct in the PCLB2-MPS2 cell shown in panel E? It looks like GFP-Csm4 is either degraded or mislocalized to the cytoplasm. Since Csm4 has a C-terminal transmembrane domain, I find it hard to believe that GFP-Csm4 would dissociate from the membranes of the ER/nuclear envelope. However, there is precedent that KASH proteins (e.g. nesprin-1-alpha) are targeted for proteolysis by the proteasome in the absence of proper nuclear envelope targeting due to the loss of A-type lamins (Muchir et al., 2006 *Biochem Biophys Res Commun*). Perhaps the authors could provide an explanation for their interesting results?
- d. It would also be helpful if the authors were to quantify the effect of their mutants on nuclear area/shape/volume.

8) Figure 3:

- a. The authors should indicate which part of the Merged images shown in panel A were used to generate the 2x zoom images.
- b. I would very much like to see some quantification of colocalization for the results presented in this figure.
- c. The inclusion of arrows to draw the reader's attention to specific colocalizations would also be useful.

9) Figure 4:

- a. Do the authors think that there is any significance to the difference observed across the three strains at 2 hours after induction of meiosis shown in panel C? In the absence of statistical tests, it is hard for me to tell.
- b. Panel A: I'd appreciate it if the authors could provide images from the separate color channels in addition to the merged images.

10) Figure 5:

- a. Panels B, C, D: I'd appreciate it if the authors could provide images from the separate color channels in addition to the merged images.
- b. Panels B and C: The schematic diagrams provided need to be better labeled. For example, it would be helpful if the authors could indicate where the nucleoplasm and cytoplasm are.
- c. Panel E: In the absence of any measurements of stoichiometry, I would refrain from drawing the Csm4:Mps2 interaction as being 1:1. Also, the differently colored domains in the proteins drawn in this figure need to be explained. A key would be helpful. That being said, the way that the authors draw the Mps2-Csm4 interaction makes it seem like these proteins interact via their coiled-coil domains. Again, in the absence of any experimental evidence I would refrain from committing to this level of detail in this model.

11) Figure S2:

- a. Panel A: I'd appreciate it if the authors could provide images from the separate color channels in addition to the merged images.

12) Figure S3:

- a. Panel E: I'd like to see images of the actin cytoskeleton in cells treated with DMSO or LatB to control for the effectiveness of actin depolymerization under these experimental conditions.

13) Figure S4:

- a. Panel B: Some quantification of these results would be much appreciated.

14) Figure S5:

- a. Panel B: Some quantification of these results would be much appreciated.

Minor Issues

- 1) I have a problem with calling a LINC complex simply a LINC. The abbreviation "LINC" should really always be followed by the word "complex". There are multiple examples of this throughout the manuscript.
- 2) In the first paragraph of the Introduction, some of the citations provided for the sentence that starts, "The canonical LINC complex is composed of" are inappropriate. The papers that really need to be referenced here are: Starr et al., 2001 Development and Crisp et al., 2005 J Cell Biol.
- 3) Regarding the same sentence from Minor Issue 1, the authors state, "The canonical LINC complex is composed of a pair of transmembrane proteins". I would recommend that the authors remove the "a pair of" from this sentence, as it makes it seem like LINC complexes are heterodimers, which is incorrect.
- 4) In the first paragraph of the Introduction, the authors state, "With SUN-KASH interaction taking place in the perinuclear space". This statement makes it seem like SUN and KASH proteins only interact within the perinuclear space, which is not entirely correct. For example, there are examples of SUN and KASH protein interactions occurring within the nucleoplasm.
- 5) Regarding the same sentence from Minor Issue 3, the authors should change the word "nucleoplasm" to "nucleoskeleton and chromatin".
- 6) Some of the citations provided by the authors for the statement that starts "LINC proteins are

believed to form heterodimeric hexamers" found at the end of the first paragraph of the Introduction are inappropriate. For example, the review by Chang et al. (2015) does not really work here.

7) The last statement of the first paragraph of the Introduction states "the stoichiometry of how they are assembled in vivo remains to be further determined". While I agree with this statement, I think that the authors need to mention that there has been a flurry of recent work where fluorescence fluctuation spectroscopy was used to quantify the stoichiometry assembly states of LINC complex proteins within the nuclear envelopes of living cells, including Hennen et al., 2017 Biophys J; Hennen and Saunders et al., 2018 Mol Biol Cell; Hennen et al., 2019 Biophys J; and Hennen and Hur et al., 2019 Methods.

8) In the second paragraph of the Results section "Meiotic Mps2 is a major binding partner of Csm4", the authors state, "we did not observe a direct Csm4-Mps3 interaction". They should remove the word "direct", as TAP of a protein and its binding partners from cell lysates will never be able to discriminate between a direct or indirect interaction.

9) In the first paragraph of the Results section "Mps2 is required for meiotic cell progression", the authors state, "Mps2 was preferentially associated with the newly duplicated SPB, which displayed a weaker Tub4-mApple signal...due to a slower fluorescence maturation time of mApple than that of GFP". However, I do not think that they authors can really make this statement without experimental evidence. It is more likely that the weaker Tub4-mApple signal observed on the newly duplicated daughter SPB is the result of the fact that the mother SPB is probably more active for microtubule polymerization.

10) Is the Tub4-mApple construct used in this work really a C-terminal fusion? Typically, XFP's are fused to the N-terminus of tubulin.

11) In the Results section "Mps2 is required for nuclear localization of Csm4 but not for Mps3", the authors state, "Mps2 and Mps3 remained bound to buy were distributed evenly around the nuclear periphery in *csm4Δ* cells...". Since these are transmembrane domain-containing proteins, the word "bound" should be changed to "localized to the nuclear envelope".

12) In the last paragraph of the Results section, the authors should include the following references for the statement, "On the basis of the current understanding of the oligomerization state of SUN and KASH proteins": Hennen et al., 2017 Biophys J; Hennen and Saunders et al., 2018 Mol Biol Cell; Hennen et al., 2019 Biophys J; and Hennen and Hur et al., 2019 Methods.

13) The authors need to indicate the light source used for their live-cell fluorescence microscopy in the Materials and Methods section.

14) In the legend for Figure 1, the authors state, "Protein structures of Csm4, Mps2, and Mps3 are shown at the bottom". However, these are illustrations of protein domain organization not structures.

Reviewer #3 (Comments to the Authors (Required)):

In this manuscript Jinbo Fan et al., report that the two KASH proteins in yeast, Mps2 and Csm4 heterodimerize, and bind together with the SUN protein Mps3, in the context of budding yeast meiosis. The heterotrimeric complex is required and is sufficient to recruit and tether the telomers through their interaction with the SUN protein Mps3. These findings are important because heterotrimeric KASH proteins might interact with variable set of cytoskeletal proteins, which potentially regulate telomere tethering during miosis.

Generally, both the immunofluorescent and biochemical data are clean and convincing.

I have minor comments:

1. The authors should add a scheme in each of the figures (as was done in Figure 5) to explain what is observed in the distinct panels. Also the borders of the cells, as well as the nuclear borders should be indicated to help the reader in understanding the images.
2. In Figure 2D explain what are the arrows. Also in this figure there was no overlap between Mps2 and Mps3.
3. Fig 5A: I do not see the rescue of CSM4 and MPS2 they look very similar to MPS2 alone.
4. Figure 5C: the localization of the patch is not clear. Also all schemes should be in similar orientation.
5. Page 9: Changelocalization to the telomere (Fig 3D)..... (not 3C).
6. In *Drosophila*, heterodimerization between the two KASH proteins, Msp300 and Klar was also reported (Elhanany-Tamir et al., 2012). It might be a general phenomenon which potentially contribute to the ability of distinct cytoskeletal elements to influence the nucleoskeleton. The authors should discuss this option, which might be of functional importance for the activity of the LINC complex in distinct cell types.

Reviewer #1 (Comments to the Authors (Required)):

This is a beautiful paper in conception, execution and presentation. I have only minor comments.

The budding yeast meiotic "t-LINC" complex transduces information across the nuclear envelope (NE), thereby enabling treadmilling of NE-associated cytoplasmic actin fibers to drive telomere-led chromosomal motions within the nucleus. Canonically such complexes include a SUN protein, localizing to the inner nuclear membrane, and a "KASH" protein localized to the outer nuclear membrane, which interact in the peri-nuclear space. The budding yeast has a single SUN-domain protein, Mps3, but lacks a canonical KASH-domain protein. It does have two outer membrane-localized KASH-like proteins, Mps2 and Csm4 (which is meiosis-specific) but exactly how these two molecules might function has been mysterious. However, the authors noted that the structure of Csm4 precludes a KASH-like role in contacting Mps3 through the peri-nuclear space. They therefore sought to identify a missing linker molecule, without bias to possible suspects by asking which molecules co-immunoprecipitate with Csm4. The authors discovered a strong interaction of Csm4 uniquely with Mps2 and confirm the interaction between the two molecules by reciprocal co-IP. They also provide genetic evidence for an interaction via interplay between the two molecules in mitotic cells, where overproduction of (meiosis-specific) Csm4 is deleterious (presumably because it titrates Mps2) and, correspondingly, this deleterious effect is suppressed by overproduction of Mps2.

Extensive additional results show that the combined action of Mps2 and Csm4 is required to carry out the canonical functions of a KASH protein in the t-LINC complex. Specifically, yeast t-LINC is composed of Csm4, Mps2 and Mps3, where Mps2 links Csm4 and Mps3. Another meiosis-specific protein, Ndj1 is required for Mps3 localization to telomeres and thus is required for localization of the entire t-LINC complex.

Thoughtful consideration of all information leads to a specific proposition for the molecular composition of the t-LINC complex and to the suggestion that non-canonical KASH molecules, and thus non-canonical LINC complexes, may be a relatively common occurrence in a variety of situations. Thus, the presented results are likely to be of general significance. The presented results are also interesting and novel with respect to the functional roles of Mps2 (and thus the t-LINC complex).

Additional remarks:

- Csm4 and Mps2 both colocalize to the nuclear periphery during meiosis, as expected from the linker hypothesis. It is also demonstrated for the first time that Mps2 and Mps3 interact. Most importantly, no interaction of Csm4 with Mps3 is detected, in accord with the motivating expectation that such interaction is mediated by a linker molecule (which they now identify as Mps2).

- Mps2 is shown to localize both to the Spindle Pole Body (SPB) and the nuclear periphery, with the specific timing and localization to a subregion of the NE, appropriate to its inferred role. This careful documentation provides a basis for clarity of understanding of dual roles for Mps2 in the two different locations.

- The authors explore the functional significance of Mps2 in a series of rigorous and thoughtful experiments. Since Mps2 has roles also in mitotic cells, they generated and analyzed a pCLB2-MPS2 allele in which meiotic expression is specifically suppressed. Careful cytology directly demonstrates a defect in SPB separation, primarily at MII but also significantly at MI, and appropriate genetic analyses imply that this is a consequence of a direct role of Mps2 at the

SPB, rather than any defect in chromosomal processes. The authors do not comment on the fact that exit from MI, even in cells that ultimately execute this division, is delayed and that this delay is eliminated by a *spo11* mutation. This is worth noting because it suggests that absence of movement results in recombination-related defects which trigger delayed onset of MI, irrespective of SPB-related failures of MI/MII to occur at all.

- Interdependency of t-LINC components for their roles in t-LINC nuclear envelope localization was carefully examined. (a) *Mps2* and *Mps3* colocalize to a portion of the nuclear periphery. (b). An especially interesting, and elegant finding is that *Mps2*, *Mps3*, *Csm4* are at leading ends of NE protrusions, which are known to be mediated by telomere-LINC-actin associations. These findings provide direct evidence for localization of these molecules at the appropriate telomere/NE/actin associations. (c) In absence of *Csm4*, *Mps2/Mps3* are still on nuclear periphery but are distributed all around, rather than in a partial domain, and no protrusions occur. Thus, *Csm4* is downstream of *Mps2/3* for actin-mediated effects. (d) In absence of *Mps2*, *Mps3* still on nuclear periphery but other effects of *csm4D* absent. Thus, *Mps2* is downstream of *Mps3*. (e) In absence of *Mps3*, *Mps2* localizes aberrantly to the NE (and so there are no nuclear protrusions). (f). Overall: *Mps3*>*Mps2*>NE localization; and > *Csm4* > NE position bias and protrusions.

- Protrusion data implying colocalization of *Mps2* to telomere/NE associations (above) is confirmed by direct analysis of telomeres in spreads. This was very rigorously done using *Rec8* as axis marker and *Ndj1* as telomere marker. In addition, this assay was used to define functional dependencies for *Mps2* telomere association specifically. Telomere association does not require *Csm4*, the outer membrane protein but does require both *Mps3* and *Ndj1*. Since *Ndj1* is required for *Mps3* localization to telomeres, which is required for *Mps2* localization to telomeres, and in accord with the idea that *Mps2* links *Mps3* and *Csm4*, depletion of *Mps2* did not affect *Mps3* telomere localization but abolished *Csm4* localization.

The authors have also carefully defined the functional roles of *Mps2* for meiotic chromosomes. (i) They confirm the expectation that *Mps2* is required for polarized localization of telomeres to a subdomain of the nuclear envelope (the "bouquet"). They also provide nice evidence that this stage is transient. (ii) They show that close juxtaposition of tagged chromosomal loci ("pairing") occurs as in wild type but that this process is delayed. To this reviewer's eye, this delay comes "late" in prophase, perhaps implying a problem in interlock resolution rather than a problem in pairing per se. This might merit some discussion. (iii) They show that the frequency of heteroallelic recombination, tested at two loci, is reduced 10x. This is very interesting and novel and the explanation is not obvious. It might be due to a lack of pairing; however, there are also more complicated explanations that the authors might wish to consider, particularly in light of the fact that tracts of heteroduplex DNA (which underlie heteroallelic recombination) are longer for interactions that give crossovers than for interactions that give noncrossovers.

The authors next present a lovely set of experiments in which they reconstruct t-LINC activity in vegetatively growing (mitotically dividing) cells. They express the two meiosis-specific components (*Ndj1* and *Csm4*) in vegetative cells. *Mps2* and *Mps3* are present in such cells as well as in meiosis. They discover that telomeres, marked by *Rap1-GFP*, enter daughter cells precociously, prior to SPB separation, as seen previously for *Mps3*. Thus, the t-LINC complex can move telomeres in vegetative cells without addition of any other meiosis-specific components.

The authors draw on published structural information to propose that the yeast t-LINC is a nonamer (Fig 5E). They point clear analogies between the yeast *Csm4/Mps2* collaboration with molecules in *Arabidopsis* and point out that numerous KASH variants also exist in metazoans. They suggest that variant LINC complexes could be prevalent and thus that the current study can provide insight into LINC assembly and its evolution in eukaryotes.

The authors appropriately note that a paper published in April of this year also suggests that Mps2 acts as a component of t-LINC. The two studies have obviously been carried out in parallel. This reviewer finds the current work to be the much more elegant and complete of the two studies and to be a powerful extension to the other study. The current work uses an unbiased approach to identify Mps2 as a relevant component rather than targeting it specifically; provides much more complete characterization of the localization and effects of Mps2 and t-LINC and of Mps2 interactions with other components; provides uniquely important functional analysis; and uniquely considers the broader implications of the findings for non-canonical KASH-related proteins.

In summary, the current paper is an important, and carefully- and exhaustively-executed contribution to our basic understanding of the budding yeast meiotic t-LINC complex in particular with implications for LINC-mediated complexes in general.

Our response to reviewer #1:

i. As the reviewer suggested, we have now addressed and discussed the phenotype of delayed SPB separation in *P_{CLB2}-MPS2* cells (page 7 bottom paragraph and page 8, lines1-6)

ii. We have discussed Mps2's role in homolog pairing and interlock resolution (page 11, second paragraph).

Reviewer #2 (Comments to the Authors (Required)):

This manuscript investigates the assembly and function of LINC complexes in budding yeast. Specifically, it asks how the KASH-like protein Csm4 interacts with the SUN protein Mps3 within the perinuclear space to promote the formation of a telomere-associated LINC (t-LINC) complex. This question is particularly interesting, given the lack of a clear canonical SUN protein-interacting KASH peptide at the C-terminus of Csm4. The authors provide evidence to suggest a model where telomere-associated Mps3 indirectly interacts with Csm4 through Mps2 to form a so-called "heterotrimeric" t-LINC complex, which is required for proper telomere movement and meiotic recombination. In addition, the authors show that expressing the normally meiotic CSM4 in vegetative cells results in the reconstitution of the t-LINC complex that is capable of tethering telomeres to the nuclear envelope. Overall, this intriguing work lays the foundation for understanding the important question of how LINC complexes composed of different KASH proteins are assembled to perform their differential functions in cells. However, before I can recommend this manuscript for publication, the authors need to address the following major and minor issues.

Major Issues

1) A major issue that I have with this manuscript is its improper description of the stoichiometry of the LINC complex. For example, the word "heterotrimeric" does not really work to explain the stoichiometry of the t-LINC complex composed of Csm4, Mps2, and Mps3. Based on the literature, what can be said is that Mps3 likely forms a homo-trimer and that a three Mps2 proteins can associate with a Mps3 homo-trimer, resulting in the formation of a hetero-hexameric Mps2-Mps3 LINC complex. Since the stoichiometry of the Csm4-Mps2 interaction is currently unknown, the authors cannot conclude that the yeast t-LINC complex is a nonamer, as they do in the last paragraph of the Results section. Therefore, I would strongly caution against the use of precise stoichiometry to describe t-LINC complexes containing Csm4-Mps2-Mps3.

Our response:

We appreciate the reviewer's comments on the stoichiometry between Csm4 and Mps2. On the basis of (i) our finding that Csm4 and Mps2 interact physically (Fig 1B and 1C) and form oligomers (our unpublished data), (ii) Mps2 is thought to form a homo-oligomer through its coiled-coil regions (Zizlsperger and Keating, 2009, J Structural Biology), (iii) sequence similarity between Csm4 and Mps2 is limited to their coiled-coil regions (CC of Csm4, CC2 of Mps2. Fig 1A), and (iv) Bbp1, which competes with Csm4 (our unpublished data), binds to Mps2 at the coiled coils (Kupke et al., 2017, JBC), we provide this speculative model of the t-LINC complex as a nonamer. We have revised our discussion to stress the speculative nature of this model and to also include the alternatives (page 14, top paragraph).

2) It is unclear to me if the fusion proteins used in this work were functional or not. Were these fusions previously characterized? If so, the authors should make this clearer.

Our response:

All of our fusion protein constructs were incorporated at their endogenous loci and served as the only functional copy of the corresponding gene, i.e. there was no additional copy of the corresponding gene in the yeast genome. The following alleles have been reported previously: *MPS3-V5*, *ndt80Δ*, *MPS3-mApple*, *TUB4-mApple*, *NDJ1-V5*, *HTA1-mApple*, *REC8-GFP*, *P_{CLB2}-MPS3*, *ndj1Δ*, *RAP1-GFP* and *MPS3-GFP* (Li et al., 2014; Li et al., 2015).

Fusion proteins generated from this study are functional on the basis of their effect on cell cycle progression, tetrad formation, and spore viability. We have revised the Methods and Materials section to clarify this statement. (page 15, first and second paragraphs)

3) The authors need to better describe the results of the recently published work by Lee et al. (2020). In particular, they need to explain the current thoughts of how the t-LINC complex attaches telomeres to the actin cytoskeleton. By directly addressing the similarities and differences between their work and the work presented in Lee et al., 2020 Curr Biol, the authors will help explain their contribution to our understanding of t-LINC complex assembly and function.

Our response:

We have revised our discussion to address the concerns. For example, in our model, we have stated that alternative interactions may occur at the yeast t-LINC complex. We hypothesize that Csm4 mediates a direct interaction with the actin filament and its associated motor proteins, for example Myo2, instead of serving as an associated factor that regulates Mps2's binding to Myo2, as suggested by Lee et al. Considering that Mps2 and Mps3 are members of both the c-LINC and t-LINC complexes, we have now discussed a potential crosstalk between these two complexes in budding yeast (page 14, top paragraph).

4) How do the authors envision that Csm4 works together with Mps2-Mps3 to move telomeres? Is it that Csm4 promotes the assembly and function of Mps2-Mps3 LINC complexes? Does Mps2 still interact with Mps3 in cells lacking Csm4? Alternatively, does Csm4 regulate the interaction of Mps2 with Myo2, as suggested by Lee et al., 2020 Curr Biol?

Our response:

See our response above to point#3

5) Figures 2C, 4B, 4C, 4D, 4E, S1B, S1C, S1D, S1E, S1F, S1G, and S2D: The authors show plots of representative experiments and state that they performed 3 biological replicates. I would prefer it if the authors could report a plot of the average measurements from each of the 3 replicates. This would also allow them to perform statistical analyses of these results.

Our response:

In the figures mentioned by this reviewer, shown are the results of a representative experiment, where the control and treatment(s) were sampled simultaneously. At each time point, more than 100 cells were sampled to determine cell cycle progression on the basis of SPB separation. For meiotic studies, we used the SK1 genetic background and a standard protocol for synchronization, the strategy of which permits us to synchronize yeast cells in meiosis. Because cell synchrony is not even close to 100%, in our view, averaging the meiotic time course data can be misleading.

6) Figure 1:

- a. There is a general lack of controls for the TAP experiments presented in this work. Without these controls, it is difficult for me to assess their results.
- b. Why is there a doublet in the Anti-TAP blot for the TAP-CSM4/MPS3-3HA experiment shown in panel C?
- c. It is very hard to see the Mps3-V5 bands in the TAP-MPS2/MPS3-V5 experiment shown in panel D.
- d. How many times were these experiments replicated?
- e. What are the sizes of the z-steps shown in panel E? I am not sure that much is gained from showing these z-sections.
- f. I would like the authors to do the following:
 - i. Perform a structure/function analysis of the Mps2-Csm4 interaction to identify how these proteins associate with each other. Since both proteins have coiled-coil domains, are these required for their interaction?
 - ii. If the authors' hypothesis that Csm4 indirectly interacts with Mps3 via Mps2 were correct, I would anticipate that Mps3 would not be able to immunoprecipitate Csm4 in cells lacking Mps2. Nor would Csm4 be able to immunoprecipitate Mps3 from Mps2-null cells. The authors really need to do these experiments to substantiate their model.
 - iii. Similar to point ii above, images of the nuclear envelope localization of Csm4 in Mps3-null and Mps2-null cells would be helpful.

Our response:

- (a) In the four immunoprecipitation experiments shown in Figs 1C and 1D, three different TAP fusion proteins (TAP-Mps2, TAP-Csm4, and TAP-Mps3) were used. We would argue they served as positive controls to each other. The level of Pgk1 served as the negative control.
- (b) The double band in Panel C may represent a degradation product.
- (c) We have adjusted contrast to show the Mps3-V5 band.
- (d) In Fig 1 legend, we have now stated the number of biological replicates used.
- (e) These individual z-sections permitted us to show the location of the SPB, indicated by the arrows.
- (f) (i) This is a good suggestion, we have discussed the interaction between Csm4 and Mps2 potentially through their coiled-coil regions, also see model in Fig 7. (ii) We have stated that even in wild-type cells, physical interaction between Csm4 and Mps3 was not detected by TAP-based methods, IF and mass-spec, indicating their interaction is either indirect or weak (page 6, lines 16-18). This result and others prompted us to hypothesize

that a linker protein is needed to connect Csm4 to Mps3. (iii) Csm4 localization in a *mps2* null background is shown Fig 2E.

7) Figure 2:

- a. Quantification of the colocalization of Mps3-mApple with GFP-Mps2 (panel D) or GFP-Csm4 (panel E) would help the authors strengthen their conclusions.
- b. Why does the nuclear envelope look so ratty in the WT cell that express GFP-Csm4 and Mps3-mApple (panel E) compared to nuclear envelope of the WT cell that expresses GFP-Mps2 and Mps3-mApple (panel D)?
- c. What happened to the GFP-Csm4 construct in the PCLB2-MPS2 cell shown in panel E? It looks like GFP-Csm4 is either degraded or mislocalized to the cytoplasm. Since Csm4 has a C-terminal transmembrane domain, I find it hard to believe that GFP-Csm4 would dissociate from the membranes of the ER/nuclear envelope. However, there is precedent that KASH proteins (e.g. nesprin-1-alpha) are targeted for proteolysis by the proteasome in the absence of proper nuclear envelope targeting due to the loss of A-type lamins (Muchir et al., 2006 Biochem Biophys Res Commun). Perhaps the authors could provide an explanation for their interesting results?
- d. It would also be helpful if the authors were to quantify the effect of their mutants on nuclear area/shape/volume.

Our response:

- a. **Mps2 and Mps3 colocalize with each other along the nuclear periphery and at the SPB, Mps3 colocalizes with Csm4 at the nuclear periphery, but not at the SPB. We have now noted that some Mps3 foci at the nuclear periphery did not colocalize with Mps2, indicating that Mps3 forms protein aggregates outside of the context of the LINC complex (page 8, lines17-19).**
- b. **These cells at prophase I show prominent nuclear protrusions, i.e. regions indicated by the arrows. We have now included in Fig 2F our quantification of nuclear shape.**
- c. **In the absence of Mps2, Csm4 shows a diffused localization. Our interpretation is that in the absence of Mps2, Csm4 becomes mislocalized, e.g. Csm4 is no longer preferentially associated with the nuclear envelope. We have now presented two potential outcomes, as the review suggested, in which Csm4 either diffuses throughout the cytoplasm or is degraded. Our unpublished data suggests Csm4 is subject to degradation by the ubiquitin proteasome. (page 9, lines6-8)**
- d. **We have now included the nuclear shape data in Fig 2F.**

8) Figure 3:

- a. The authors should indicate which part of the Merged images shown in panel A were used to generate the 2x zoom images.
- b. I would very much like to see some quantification of colocalization for the results presented in this figure.
- c. The inclusion of arrows to draw the reader's attention to specific colocalizations would also be useful.

Our response:

- a. **We have now included frames to indicate the enlarged views**
- b. **We have stated that “each Ndj1 focus is associated with a corresponding Mps2 focus in the cell shown in Fig 3A” (page 10, lines5-6).**
- c. **We have added a note in Fig 3C legend to draw attention of “telomere” localization of Mps2 in *WT* and *csm4Δ* cells.**

9) Figure 4:

- a. Do the authors think that there is any significance to the difference observed across the three strains at 2 hours after induction of meiosis shown in panel C? In the absence of statistical tests, it is hard for me to tell.
- b. Panel A: I'd appreciate it if the authors could provide images from the separate color channels in addition to the merged images.

Our response:

- a. We agree there were variations at 2h. The data in Panel C show cell viability, which is used as a denominator for calculating the rate of gene conversion. As such, the variations at 2h did not affect the overall result and our interpretation.**
- b. We have included the single channel images from Rap1-GFP, the other set of images is for marking the SPB, which basically shows an SPB focus. In our view, the SPB is well represented in the merged images.**

10) Figure 5:

- a. Panels B, C, D: I'd appreciate it if the authors could provide images from the separate color channels in addition to the merged images.
- b. Panels B and C: The schematic diagrams provided need to be better labeled. For example, it would be helpful if the authors could indicate where the nucleoplasm and cytoplasm are.
- c. Panel E: In the absence of any measurements of stoichiometry, I would refrain from drawing the Csm4:Mps2 interaction as being 1:1. Also, the differently colored domains in the proteins drawn in this figure need to be explained. A key would be helpful. That being said, the way that the authors draw the Mps2-Csm4 interaction makes it seem like these proteins interact via their coiled-coil domains. Again, in the absence of any experimental evidence I would refrain from committing to this level of detail in this model.

Our response:

- a. The original Figure 5 is now Figure 6, in which we have included single channel images.**
- b. We have revised the diagrams with additional labels in Figure 6A and 6B**
- c. Our model is now included in Figure 7. We have discussed the rationale of our model, also see our response above to points #1 and #3.**

11) Figure S2:

- a. Panel A: I'd appreciate it if the authors could provide images from the separate color channels in addition to the merged images.

Our response:

We have included the single channel images of GFP-Mps2. Tub4-mApple marks the SPB, which shows a dot.

12) Figure S3:

- a. Panel E: I'd like to see images of the actin cytoskeleton in cells treated with DMSO or LatB to control for the effectiveness of actin depolymerization under these experimental conditions.

Our response:

We used a published protocol to depolymerize actin filaments with Latriculin B (Koszul et al., 2008, Cell, 133:1188-1201). Although we did not directly observe actin filaments, on

the basis of our observed Mps3 patch formation, actin depolymerization appeared effective in treated cells.

13) Figure S4:

a. Panel B: Some quantification of these results would be much appreciated.

Our response:

We have now included our quantification in new Fig S3B to address this concern.

14) Figure S5:

a. Panel B: Some quantification of these results would be much appreciated.

Our response:

In both the raffinose and galactose treatments, the percentage of cells forming the Mps3 patch was essentially zero, we have included this information in the text (page 12, lines 19-20).

Minor Issues

1) I have a problem with calling a LINC complex simply a LINC. The abbreviation "LINC" should really always be followed by the word "complex". There are multiple examples of this throughout the manuscript.

Our response:

We have revised the term “t-LINC complex” wherever is appropriate.

2) In the first paragraph of the Introduction, some of the citations provided for the sentence that starts, "The canonical LINC complex is composed of" are inappropriate. The papers that really need to be referenced here are: Starr et al., 2001 Development and Crisp et al., 2005 J Cell Biol.

Our response:

We have revised as suggested. Page 3, lines 7-8.

3) Regarding the same sentence from Minor Issue 1, the authors state, "The canonical LINC complex is composed of a pair of transmembrane proteins". I would recommend that the authors remove the "a pair of" from this sentence, as it makes it seem like LINC complexes are hetero-dimers, which is incorrect.

Our response:

We have revised the wording in the abstract and in the introduction.

4) In the first paragraph of the Introduction, the authors state, "With SUN-KASH interaction taking place in the perinuclear space". This statement makes it seem like SUN and KASH proteins only interact within the perinuclear space, which is not entirely correct. For example, there are examples of SUN and KASH protein interactions occurring within the nucleoplasm.

Our response:

Revised to “canonical SUN-KASH” interaction. Page 3, line 8.

5) Regarding the same sentence from Minor Issue 3, the authors should change the word "nucleoplasm" to "nucleoskeleton and chromatin".

Our response:

We have revised this sentence as suggested.

6) Some of the citations provided by the authors for the statement that starts "LINC proteins are believed to form heterodimeric hexamers" found at the end of the first paragraph of the Introduction are inappropriate. For example, the review by Chang et al. (2015) does not really work here.

Our response:

We have removed the "Chang et al." citation.

7) The last statement of the first paragraph of the Introduction states "the stoichiometry of how they are assembled in vivo remains to be further determined". While I agree with this statement, I think that the authors need to mention that there has been a flurry of recent work where fluorescence fluctuation spectroscopy was used to quantify the stoichiometry assembly states of LINC complex proteins within the nuclear envelopes of living cells, including Hennen et al., 2017 Biophys J; Hennen and Saunders et al., 2018 Mol Biol Cell; Hennen et al., 2019 Biophys J; and Hennen and Hur et al., 2019 Methods.

Our response:

We have revised this sentence.

8) In the second paragraph of the Results section "Meiotic Mps2 is a major binding partner of Csm4", the authors state, "we did not observe a direct Csm4-Mps3 interaction". They should remove the word "direct", as TAP of a protein and its binding partners from cell lysates will never be able to discriminate between a direct or indirect interaction.

Our response:

Revised to "physical interaction"

9) In the first paragraph of the Results section "Mps2 is required for meiotic cell progression", the authors state, "Mps2 was preferentially associated with the newly duplicated SPB, which displayed a weaker Tub4-mApple signal...due to a slower fluorescence maturation time of mApple than that of GFP". However, I do not think that they authors can really make this statement without experimental evidence. It is more likely that the weaker Tub4-mApple signal observed on the newly duplicated daughter SPB is the result of the fact that the mother SPB is probably more active for microtubule polymerization.

Our response:

We prefer our interpretation

10) Is the Tub4-mApple construct used in this work really a C-terminal fusion? Typically, XFP's are fused to the N-terminus of tubulin.

Our response:

Tub4-mApple is C-terminally tagged. We have reported this construct previous, e.g. Shirk et al., 2011 JCS, Li et al., 2015 JCB.

11) In the Results section "Mps2 is required for nuclear localization of Csm4 but not for Mps3",

the authors state, "Mps2 and Mps3 remained bound to buy were distributed evenly around the nuclear periphery in *csm4Δ* cells...". Since these are transmembrane domain-containing proteins, the word "bound" should be changed to "localized to the nuclear envelope".

Our response:
Revised as suggested.

12) In the last paragraph of the Results section, the authors should include the following references for the statement, "On the basis of the current understanding of the oligomerization state of SUN and KASH proteins": Hennen et al., 2017 Biophys J; Hennen and Saunders et al., 2018 Mol Biol Cell; Hennen et al., 2019 Biophys J; and Hennen and Hur et al., 2019 Methods.

Our response:
We have removed this sentence.

13) The authors need to indicate the light source used for their live-cell fluorescence microscopy in the Materials and Methods section.

Our response:
We have included this information in the revised manuscript (page 18, bottom paragraph).

14) In the legend for Figure 1, the authors state, "Protein structures of Csm4, Mps2, and Mps3 are shown at the bottom". However, these are illustrations of protein domain organization not structures.

Our response:
We have revised the sentence to: "Domain organization of Csm4, Mps2 and Mps3 is shown at the bottom."

Reviewer #3 (Comments to the Authors (Required)):

In this manuscript Jinbo Fan et al., report that the two KASH proteins in yeast, Mps2 and Csm4 heterodimerize, and bind together with the SUN protein Mps3, in the context of budding yeast meiosis. The heterotrimeric complex is required and is sufficient to recruit and tether the telomers through their interaction with the SUN protein Mps3. These findings are important because heterotrimeric KASH proteins might interact with variable set of cytoskeletal proteins, which potentially regulate telomere tethering during miosis. Generally, both the immunofluorescent and biochemical data are clean and convincing.

I have minor comments:

1. The authors should add a scheme in each of the figures (as was done in Figure 5) to explain what is observed in the distinct panels. Also the borders of the cells, as well as the nuclear borders should be indicated to help the reader in understanding the images.

Our response:
We have revised the followings to improve our figures:
i. Added cell outline in Fig 1E
ii. Inserted frames in Fig 3 to indicate the enlarged regions
iii. Revised diagram labels in Fig 5 (now Fig 6)

2. In Figure 2D explain what are the arrows. Also in this figure there was no overlap between Mps2 and Mps3.

Our response:

Arrows in Fig 2D point to the nuclear protrusion formed by the t-LINC complex. This information is included in Fig 2D legend. We have also revised our description of Mps2 and Mps3 colocalization. (page 8, bottom paragraph)

3. Fig 5A: I do not see the rescue of CSM4 and MPS2 they look very similar to MPS2 alone.

Our response

We have revised our interpretation of this result: overexpression of *MPS2* suppressed the *csm4* mutant phenotype in the galactose medium. (page 12, lines 4-6)

4. Figure 5C: the localization of the patch is not clear. Also all schemes should be in similar orientation.

Our response:

We have included the single channel images in Fig 6 to show the Mps3 patch (originally Fig 5)

5. Page 9: Changelocalization to the telomere (Fig 3D)..... (not 3C).

Our response:

Revised as suggested.

6. In *Drosophila*, heterodimerization between the two KASH proteins, Msp300 and Klar was also reported (Elhanany-Tamir et al., 2012). It might be a general phenomenon which potentially contribute to the ability of distinct cytoskeletal elements to influence the nucleoskeleton. The authors should discuss this option, which might be of functional importance for the activity of the LINC complex in distinct cell types.

Our response:

We have included this citation. (page 14, second paragraph)

September 10, 2020

RE: Life Science Alliance Manuscript #LSA-2020-00824-TR

Dr. Hong-Guo Yu
Florida State University
Biological Science
89 Chieftan Way
Tallahassee, Florida 32306

Dear Dr. Yu,

Thank you for submitting your revised manuscript entitled "Mps2 links Csm4 and Mps3 to form a telomere-associated LINC complex in budding yeast". Your revised manuscript has been reviewed by referees (report appended below) and we are happy to announce that your manuscript is ready to be published in Life Science Alliance pending final revisions necessary to meet our formatting guidelines.

Along with the points listed below, please address the following as well:

- please add ORCID ID for corresponding author-you should have received instructions on how to do so
- please add a conflict of interest statement to your main manuscript text
- please add a scale bar to Fig 2D and Fig S3A
- please separate the combined Results and Discussion section into a separate Results section and a separate Discussion section

A. FINAL FILES:

B. MANUSCRIPT ORGANIZATION AND FORMATTING:

Sincerely,

Shachi Bhatt, Ph.D.
Executive Editor
Life Science Alliance

Reviewer #2 (Comments to the Authors (Required)):

Overall, the authors have successfully addressed the concerns that I raised in my review of their original manuscript. I recommend it for publication.

September 11, 2020

RE: Life Science Alliance Manuscript #LSA-2020-00824-TRR

Dr. Hong-Guo Yu
Florida State University
Biological Science
89 Chieftan Way
Tallahassee, Florida 32306

Dear Dr. Yu,

Thank you for submitting your Research Article entitled "Mps2 links Csm4 and Mps3 to form a telomere-associated LINC complex in budding yeast". It is a pleasure to let you know that your manuscript is now accepted for publication in Life Science Alliance. Congratulations on this interesting work.

DISTRIBUTION OF MATERIALS:

Again, congratulations on a very nice paper. I hope you found the review process to be constructive and are pleased with how the manuscript was handled editorially. We look forward to future exciting submissions from your lab.

Sincerely,

Shachi Bhatt, Ph.D.
Executive Editor
Life Science Alliance